# Abundant pleiotropy across neuroimaging modalities identified through a multivariate genome-wide association study

E. P. Tissink [1,2] ✉, A. A. Shadrin [3], D. van der Meer [3,4], N. Parker [3], G. Hindley [3,5], D. Roelfs [3], O. Frei [3], C. C. Fan [6,7], M. Nagel[1], T. Nærland[8], M. Budisteanu[9,10], S. Djurovic [3,8,11], L. T. Westlye [3,8,12], M. P. van den Heuvel [1,13], D. Posthuma [1,13], T. Kaufmann [3,14], A. M. Dale [7,15,16] & O. A. Andreassen [3,8] ✉

Genetic pleiotropy is abundant across spatially distributed brain characteristics derived from one neuroimaging modality (e.g. structural, functional or diffusion magnetic resonance imaging [MRI]). A better understanding of pleiotropy across modalities could inform us on the integration of brain function, micro- and macrostructure. Here we show extensive genetic overlap across neuroimaging modalities at a locus and gene level in the UK Biobank (N = 34,029) and ABCD Study (N = 8607). When jointly analysing phenotypes derived from structural, functional and diffusion MRI in a genome-wide association study (GWAS) with the Multivariate Omnibus Statistical Test (MOSTest), we boost the discovery of loci and genes beyond previously identified effects for each modality individually. Cross-modality genes are involved in fundamental biological processes and predominantly expressed during prenatal brain development. We additionally boost prediction of psychiatric disorders by conditioning independent GWAS on our multimodal multivariate GWAS. These findings shed light on the shared genetic mechanisms underlying variation in brain morphology, functional connectivity, and tissue composition.

The brain is our most complex organ, rapidly integrating information from many different sources[1], with strong genetic influences[2]. Studying genetic influences through genome-wide association studies (GWAS) has shown that most loci and genes show association with multiple traits[3], a phenomenon known as "statistical pleiotropy" (e.g. one gene influences multiple phenotypes directly, or indirectly via a causal pathway or common factor)[4]. Recently, numerous loci and genes with pleiotropic effects across brain characteristics derived from a single neuroimaging modality (e.g. structural, functional or diffusion MRI) have been discovered[5–9]. Yet the majority of pleiotropic loci act across rather than within phenotype domains[3], indicating that the genes associated within these loci may show pleiotropic effects across neuroimaging modalities, but the extent of this is underexplored. Investigating how genes influence a wide variety of brain imaging phenotypes may shed light on the mechanisms underlying

alterations in brain morphology, activity, connectivity, and tissue composition that often co-occur in heritable psychiatric disorders[10,11].

Complex traits are affected by thousands of genetic variants scattered throughout the genome which makes pleiotropy between them inevitable[12]. In presence of such abundant pleiotropy, multivariate GWAS models have greater statistical power than univariate GWAS models[13]. This is because the cumulative evidence from different phenotypes leads to more sensitive detection of genetic associations and reduces the burden of multiple testing[7]. Previous studies have investigated either structural MRI-derived (sub)cortical volumes, surface area, and thickness[5–7], functional MRI-derived brain connectivity[9], or diffusion MRI-derived brain tissue composition[8] in a multivariate GWAS framework. The identified loci and genes inform us about the biological signal that is picked up by MRI, such as linking genetic effects of diffusion MRI to synaptic pruning, neuroinflammation, and axonal growth[8], structural

MRI to neurogenesis and cell differentiation[5], and functional MRI to mental health[14] and psychiatric disorders[9]. Yet the modality specificity of these links remains unclear. Moreover, many loci are currently not discoverable in unimodal analyses: a multimodal multivariate GWAS could identify those loci that can improve our understanding of the interplay of biological processes contributing to brain structural organization on multiple scales and reveal underpinnings of the structure-function relationship from a genetics viewpoint.

A large body of neuroimaging research established associations between psychiatric disorders and brain structure[15–20], function[21,22] and diffusion metrics[23,24]. However, previous studies were not able to detect non-null genetic correlations[25–27], possibly due to mixed effect directions[28] and differential polygenicity between neuroimaging phenotypes and psychiatric disorders[29]. Alternative methods that leverage the shared genetic signal between brain morphology and psychiatric

disorders have been more fruitful[28] allowing improved prediction of disease liability[30]. Given that alterations of brain morphology, activity, connectivity, and tissue composition often co-occur in heritable psychiatric disorders[10,11], using multimodal GWAS associations may potentially further aid in leveraging the shared genetic signal between neuroimaging traits and psychiatric conditions.

Here, we demonstrate evidence of extensive pleiotropy across neuroimaging modalities using 583 structural (sMRI), resting-state functional (fMRI) or diffusion (dMRI) MRI-derived phenotypes in the UK Biobank and the Adolescent Brain Cognitive Development (ABCD) Study®. We do so by performing unimodal and multimodal multivariate GWAS with the Multivariate Omnibus Statistical Test (MOSTest), which was designed to boost statistical power by capitalizing on the distributed nature of genetic influences across imaging-derived phenotypes[7] (se Fig. 1 for an overview of the study). Specifically, we 1)

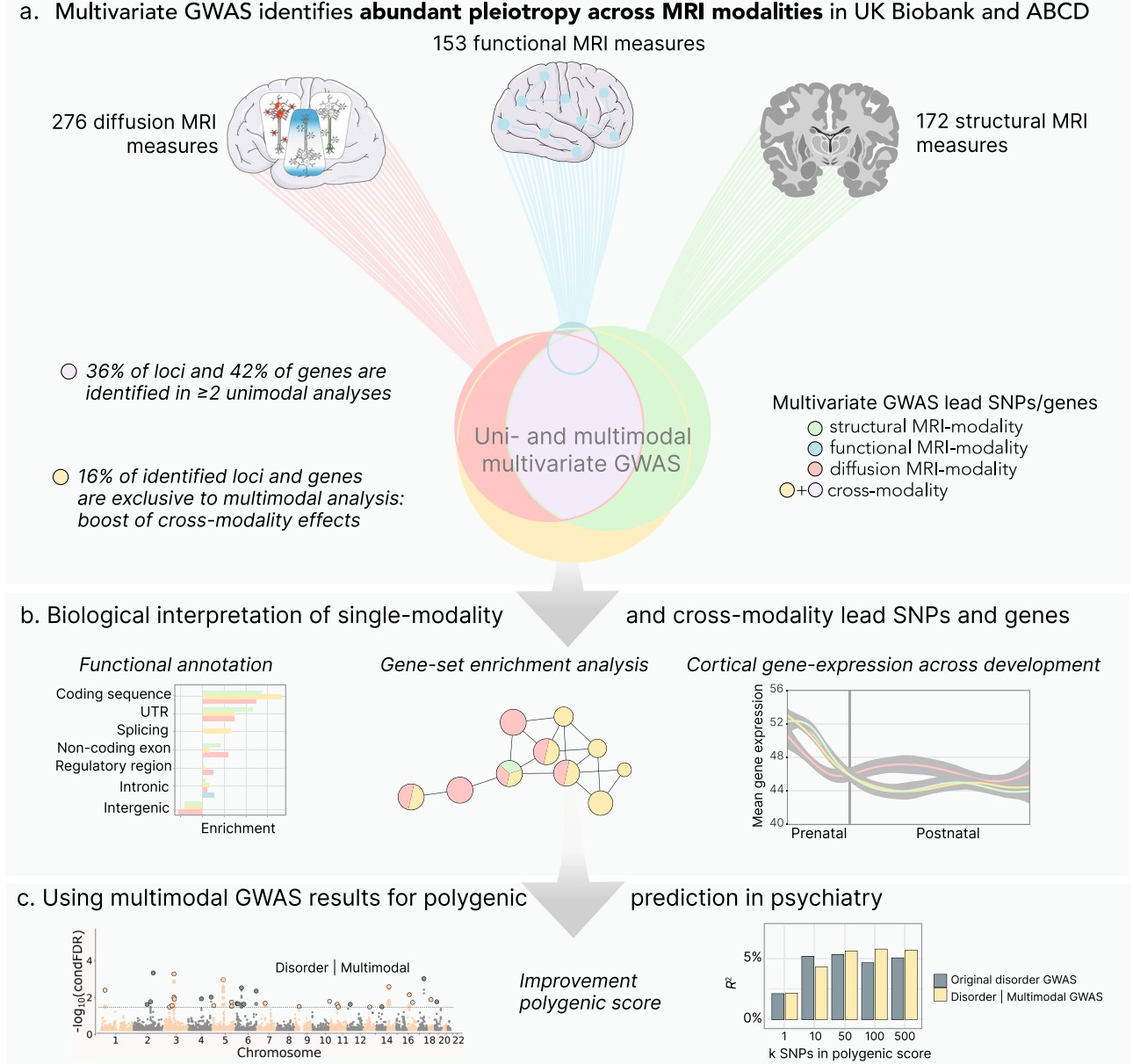

**Fig. 1 | Overview of the study. a** This study investigates statistical pleiotropy by overlapping loci and genes across unimodal multivariate GWAS, and additionally performing a multimodal multivariate GWAS. **b** Based on the overlapping patterns we classify single-modality and cross-modality loci and genes, for which we investigated potential differences in biological convergence. **c** Lastly, we use the multimodal multivariate GWAS statistics to adapt standard polygenic scores for five psychiatric conditions. This Figure includes images obtained from Servier Medical Art (https://smart.servier.com/), licensed under a Creative Commons Attribution 4.0 Unported License (https://creativecommons.org/licenses/by/4.0/).

overlap the previously described multivariate effects for each modality and 2) combine all MRI-derived phenotypes into one multimodal multivariate GWAS, obtaining an additional boost for discovery of cross-modality pleiotropic loci and genes. Next, we functionally annotate single-modality (identified in only one unimodal analysis) and cross-modality (identified in ≥2 unimodal analyses or unique to multimodal analysis) loci and genes to describe the biological signal across MRI modalities. Last, we improve polygenic prediction of bipolar disorder and ADHD after conditioning the GWASs of five major psychiatric disorders on our multimodal, multivariate, genetic signal of brain morphology, functional connectivity, and tissue composition. Thereby the current study provides insight into pleiotropic effects across neuroimaging modalities and their relevance for understanding the neurobiology of the human brain and mental health conditions.

## Results

### Abundant pleiotropy of loci and genes across neuroimaging modalities

We used data from three previous studies that applied MOSTest on single neuroimaging modality phenotypes: 172 sMRI-derived brain morphology measures (68 regional surface area and 68 regional thickness of the cerebral cortex, and 36 volumes of subcortical structures)[7], 153 fMRI-derived BOLD signal and connectivity measures (17 network variances and 136 network correlations)[9], and 276 dMRI-derived brain tissue composition principal components (65 restricted isotropic diffusion PCs, 124 restricted directional diffusion PCs, 87 normalized free water diffusion PCs)[8]. We investigated white British UK Biobank samples (as derived from both self-declared ethnic background and genetic principal component analysis) with quality-controlled genotypes and neuroimaging available ($N_{sMRI}$ = 34,029; $N_{fMRI}$ = 31,023; $N_{dMRI}$ = 30,106). We also included the ABCD cohort which had identical sMRI- and dMRI-derived measures and similar fMRI-derived measures (see Methods). The ABCD sample has a heterogeneous and admixed ancestral background. We created a replication sample within ABCD similar to our UKB discovery sample by selecting the subset of European individuals ($N_{sMRI}$ = 4794; $N_{fMRI}$ = 4132; $N_{dMRI}$ = 4418) assigned based on genetic ancestry factor[31] as defined in Methods. All quality controlled ABCD samples ($N_{sMRI}$ = 8607; $N_{fMRI}$ = 7277; $N_{dMRI}$ = 7853) served as an additional (ad)mixed ancestral replication sample. Next to genetic ancestry, ABCD samples were used to test the generalizability of our results across age.

Heritability estimates obtained in UK Biobank with linkage disequilibrium (LD) Score Regression[32] ranged from median = 5.80% (IQR = 3.09%) for fMRI-derived network variances to 28.00% (IQR = 6.92%) for sMRI-derived subcortical volumes (Supplementary Fig. 1 and Supplementary Data 1), with dMRI-derived measures ranking in-between (median = 13.70%, IQR = 9.63%). Genetic and phenotypic correlations were generally similar ($\rho(r_g, r_p)$ = 0.53, $p$ = 2.2 × 10$^{-16}$), and stronger within modalities than between modalities (Supplementary Fig. 2).

First, we combined all heritable (nominal $p < 0.05$) phenotypes derived from the same modality in multivariate GWAS analyses using MOSTest (Manhattan plots in Supplementary Fig. 4). This heritability filter was applied (as previously by Roelfs et al.[9]) because including non-heritable phenotypes into MOSTest analyses have been shown to reduce statistical power[7]. MOSTest estimates the correlation between measures from univariate GWAS (on randomly permuted genotype data) and sums the squared decorrelated $z$-values across univariate GWAS summary statistics (from the original genotype data) to integrate the effects across the measures into a multivariate test statistic[7]. These unimodal multivariate analyses identified 590, 42 and 512 genome-wide significant loci associated with sMRI, fMRI and dMRI respectively ($p < 5 × 10^{-8}/3$; Supplementary Data 2). These results are in line with earlier research (Supplementary Note 1). The number of genome-wide significant lead SNPs from UK Biobank that replicated at nominal significance ($p < 0.05$) in ABCD-based MOSTest summary

statistics differed across modalities (EUR: 24.46% sMRI, 8.70% fMRI, 23.63% dMRI), and were higher for the larger sample with (ad)mixed ancestries (42.12% sMRI, 15.38% fMRI, 35.39% dMRI). Replication rates after correcting for the number of lead SNPs tested are provided in Supplementary Data 3. Applying MAGMA[33] gene-level analyses to unimodal multivariate summary statistics identified 1620, 39 and 1453 genome-wide significant genes ($p < ((0.05/18,877)/3 =)$ 8.83 × 10$^{-7}$) for sMRI, fMRI and dMRI respectively (Supplementary Data 4).

When overlapping the identified loci ($p < 5 × 10^{-8}/3$) and genes ($p < 8.83 × 10^{-7}$) from each modality (Methods), we observed 326 loci (36.18% of total) and 1021 genes (41.79% of total) associated with at least two out of three modalities (Fig. 2a). This indicates pleiotropy across neuroimaging modalities both at the genome-wide significant locus and gene level. We replicated this pattern of overlap between sMRI and dMRI genome-wide significant loci and genes from UK Biobank in the ABCD cohort, though fMRI genome-wide significant loci did not overlap and fMRI genome-wide significant genes were not identified in ABCD (Supplementary Fig. 5, Supplementary Data 5, 6). Replication patterns were similar across ancestries, with all of the sMRI and dMRI loci identified in the European ABCD sample overlapping with the loci identified within the (ad)mixed ancestry ABCD sample. Locus definition parameters (LD cut-off and window size) did not have an effect on the observed locus overlap (Fig. 2a), as was apparent from sensitivity analyses presented in Supplementary Fig. 3.

### Multimodal GWAS boosts discovery of cross-modality loci and genes

We next investigated whether combining all sMRI, fMRI, and dMRI-derived measures in one multivariate analysis generated greater statistical power to identify significant pleiotropic loci and genes which show sub-threshold associations in each unimodal multivariate analysis (Manhattan plot in Supplementary Fig. 4). We therefore applied MOSTest across neuroimaging modalities, combining 583 phenotypes, identifying 794 genetic loci (Supplementary Data 2). The LD Score Regression intercept (1.04, SE = 0.04) indicated that the inflation in these multimodal MOSTest $p$-values (lambdaGC = 2.83) was driven by polygenicity and not population stratification or cryptic relatedness. Replication rates at nominal significance were comparable to previous MOSTest studies: EUR 26.45%, 37.55% (ad)mixed ancestries (see Supplementary Data 3 for rates after correction for the number of lead SNPs). One-hundred-thirty-six (15.09%) of these loci did not overlap with any of the loci identified in the unimodal multivariate analyses, suggesting that MOSTest leveraged the shared genetic signal across imaging modalities to boost the discovery of pleiotropic loci. Gene-based GWAS from MAGMA showed that of the 2242 genome-wide significant multimodal genes, 384 (15.72%) were not discovered for unimodal gene-based GWAS (Supplementary Data 4). We used the ABCD cohort to investigate the generalizability of our findings and observed a similar boost in multimodal discovery on a genome-wide significant locus and gene level (Supplementary Fig. 5, Supplementary Data 5, 6). Additionally, 81.82% of multimodal loci identified in the European ABCD sample were overlapping with the loci identified in the (ad)mixed ancestry ABCD sample.

We examined the univariate associations underlying multivariate associations from different parts of the Venn diagram (Fig. 2a). Figure 2c shows the univariate $z$-scores across sMRI-, fMRI-, and dMRI-derived phenotypes for ten lead SNPs representing different parts of the Venn diagram (Fig. 2a). A cluster map of univariate $z$-scores for all multimodal MOSTest lead SNPs is displayed in Supplementary Fig. 6. Alternatively, for every lead SNP identified in the MOSTest analyses presented in the Venn diagram (Fig. 2a) we extracted the minimum univariate $p$-value across all analyzed phenotypes. A lead SNP with a relatively high minimum univariate $p$-value would indicate that the signal should have been highly distributed across other measures for the variant to become genome-wide

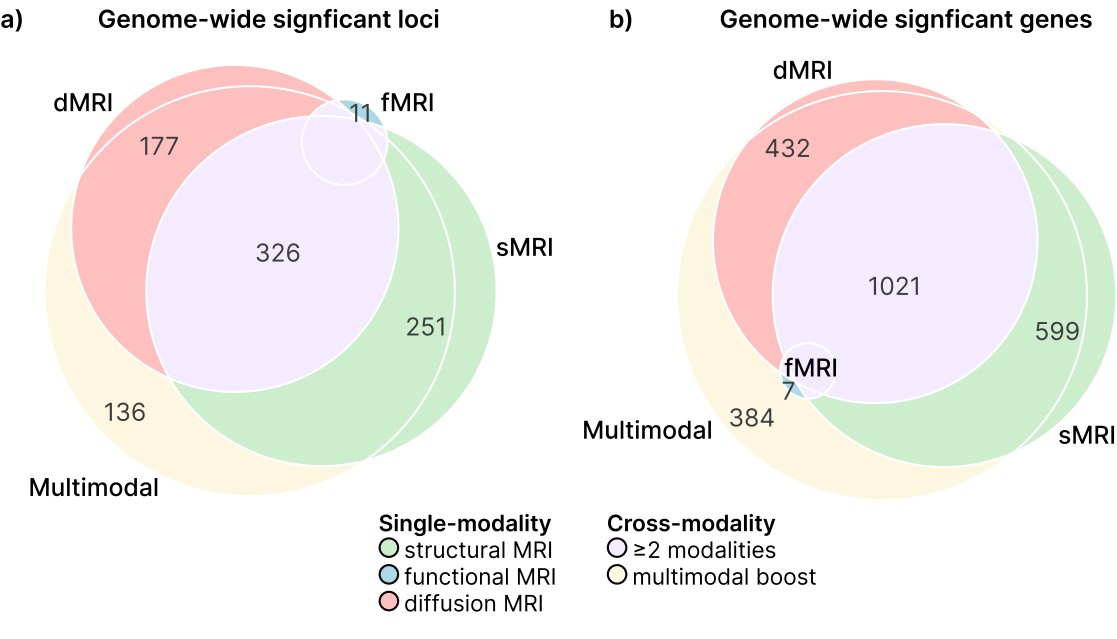

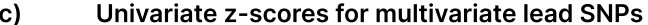

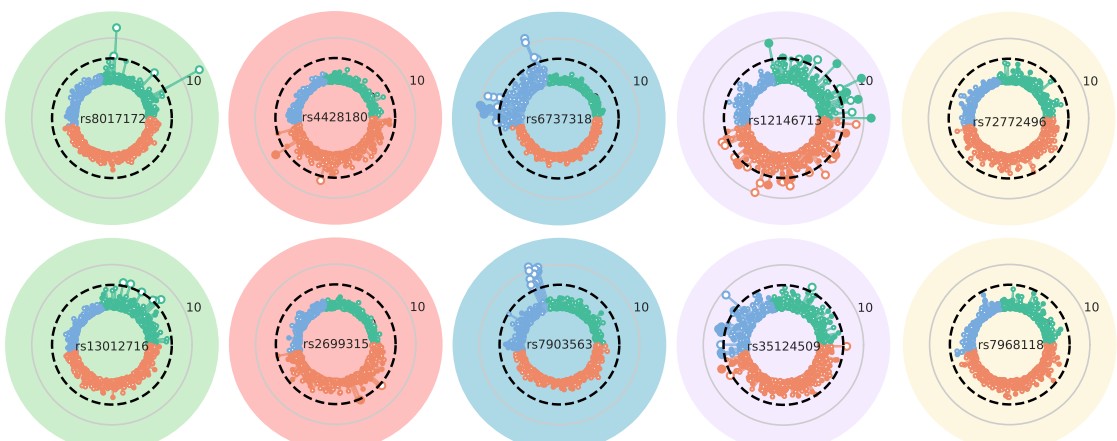

**Fig. 2 | Abundant pleiotropy of loci and genes across neuroimaging modalities.** Overlap of genome-wide significant **a** loci and **b** genes observed across neuroimaging modalities in single-modality and joint multimodal analyses with MOSTest (Bonferroni corrected $p < 5 \times 10^{-8}/3$ and $8.83 \times 10^{-7}$ respectively). When sMRI, fMRI and dMRI-derived phenotypes are jointly analyzed in MOSTest (multimodal analysis), a boost in discovery of pleiotropic loci and genes is observed (yellow). This pattern partially replicates in the ABCD cohort (Supplementary Fig. 5). For 10

example lead SNPs from different parts of a) the univariate GWAS z-scores of all phenotypes used in this study (sMRI-derived phenotypes in green, dMRI-derived phenotypes in red, fMRI-derived phenotypes in blue) are plotted in **c** as the distance from the center of each circular plot. The dashed black line corresponds to $p = 5 \times 10^{-8}$ ($z = 5.45$) and z-scores that pass this threshold are depicted larger. Positive effect direction is shown as a filled circle and negative effect direction as a white circle.

significant in the multivariate analysis. Supplementary Fig. 7 shows that lead SNPs boosted by the multimodal analysis had relatively high minimum univariate *p*-values ($0.05 > p > 5 \times 10^{-5}$) more frequently (87%) than lead SNPs of loci identified in one unimodal analysis only (sMRI-modality 72%; fMRI-modality 25%; dMRI-modality 78%), indicating that the discovery of these lead SNPs is driven by pleiotropic signals across modalities.

## Comparing characteristics of single- and cross-modality loci and genes

We investigated to what extent single-modality (identified in only one unimodal analysis) and cross-modality (identified in ≥2 unimodal analyses or boosted by the multimodal analysis) loci and genes (Supplementary Data 7, 8) differ in their biological effects compared to other complex traits. To this end, we annotated 251 sMRI-, 11 fMRI-, 177 dMRI-modality, and 462 cross-modality lead SNPs using ANNOVAR[34]

(Supplementary Data 9). We used the annotations of 43,492 (unique) lead SNPs derived from 558 traits (with reasonable power, i.e. $N > 50,000$) across 24 trait domains from Watanabe et al.[3] as a reference. Figure 3b demonstrates that cross-modality and single-modality lead SNPs were very similarly located in genomic regions as other complex traits[3,35]. Notably, cross-modality lead SNPs showed unique enrichment of exonic variants (6.58%, OR = 2.17, 95% CI = 1.43-3.17, $p = 2.91 \times 10^{-4}$). Watanabe et al. already demonstrated that the contribution of exonic SNPs increases from ~1% (SNPs associated to 1 trait domain) to ~5% (associated to ≥10 trait domain) with increasing pleiotropy[3]. The enrichment we observed therefore emphasizes the strong pleiotropic nature of the cross-modality lead SNPs.

Next, we tested whether any difference in results from gene-set enrichment analyses with Gene Ontology (GO)[36,37] biological processes, cellular components and molecular functions could be observed using the 599 sMRI-, 7 fMRI-, 432 dMRI-modality, or 1,405 cross-modality

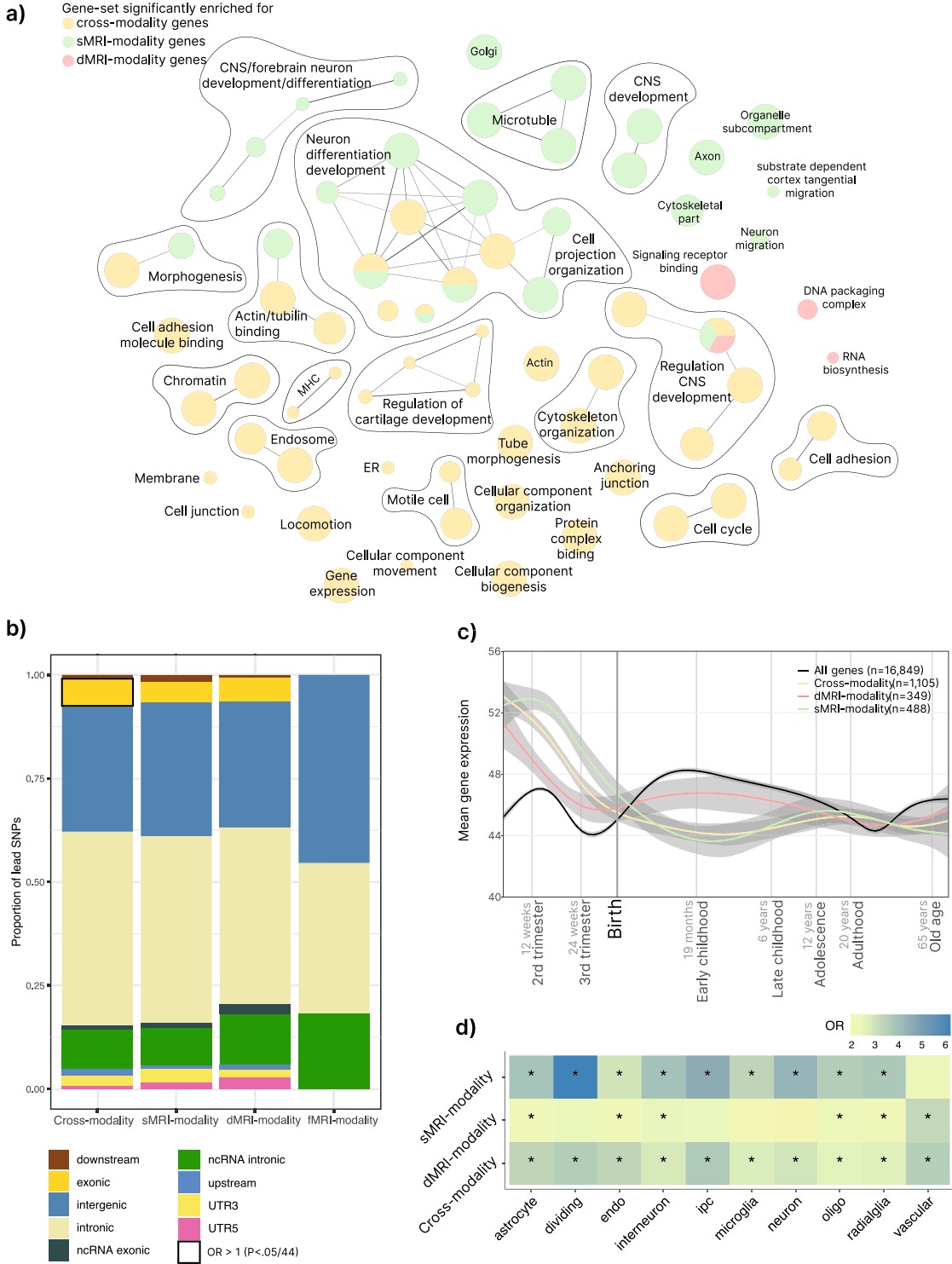

**Fig. 3 | Comparing properties of single-modality and cross-modality lead SNPs and genes. a** Gene-ontology biological processes, molecular functions and cellular components that were (Bonferroni-corrected) significantly enriched for cross-modality, sMRI-modality and/or dMRI-modality genes (none of the GO terms tested showed enrichment for the 6 fMRI-modality genes). Node size reflects gene-set size, edges reflect pathway similarity scores (Methods). **b** Functional consequences of single-modality and cross-modality lead SNPs as annotated with ANNOVAR[34]. When the null hypothesis (OR = 1) could be rejected after Bonferroni correction ($p < 1.14 \times 10^{-3}$), the solid line indicates significant enrichment of the annotation. Annotations of 43,492 (unique) lead SNPs derived from 558 traits across 24 trait

domains from Watanabe et al. were used as reference for Fisher Exact Test (Supplementary Data 9). ncRNA non-coding RNA, UTR untranslated region. **c** Mean-normalized expression (y-axis) of cross-modality and single-modality genes over developmental timepoints (x-axis; log10 scale). Gray shading indicates 95% confidence intervals. The mean-normalized expression of fMRI-modality genes is displayed in Supplementary Fig. 8, since the number of genes ($n = 5$) was low and therefore created an unreliable pattern. **d** Cell-type enrichment analysis with Fisher Exact test for fetal brain tissue from Bhaduri et al.[40]. Bonferroni corrected significant results ($p < (0.05/30 =) 1.67 \times 10^{-3}$) are indicated by an asterisk (*). OR odds ratio, ipc intermediate progenitor cells, oligo oligodendrocytes.

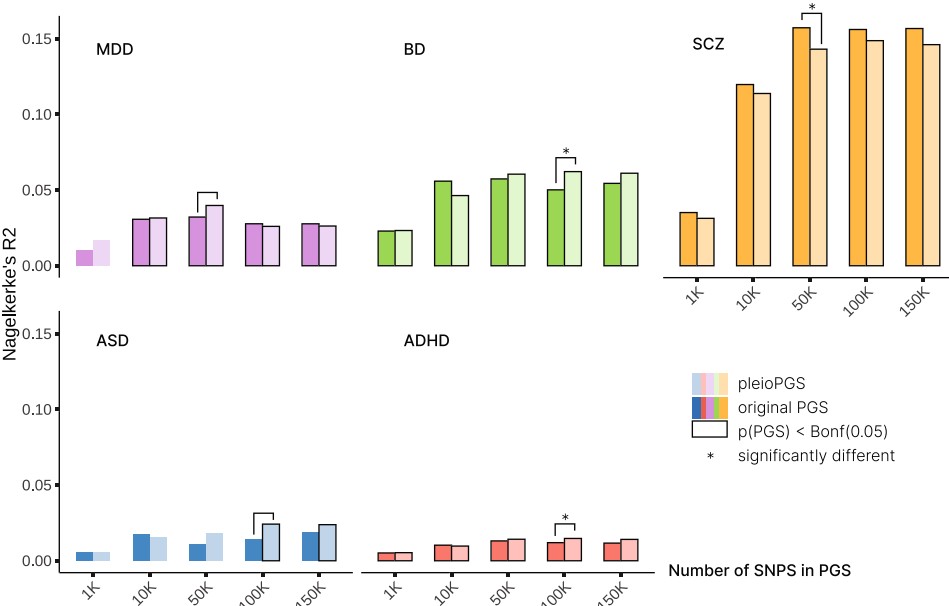

**Fig. 4 | Phenotypic variance explained for five psychiatric disorders explained by polygenic scores based on original disorder GWAS summary statistics (original PGS) and conditional disorder-multimodal GWAS summary statistics (pleioPGS).** Note that PGS using top 10 and top 100 SNPs had too low $R^2$ values to be visible and are therefore not plotted, but can be found in Supplementary Data 17. Independent target samples were used for prediction (see Methods). A likelihood ratio test was applied to compare the model which includes only the best PGS (highest $R^2$) with the model including both PGS for each disorder, and is indicated with an asterisk (*) if significant ($p < 0.05$). MDD major depressive disorder, BD bipolar disorder, SCZ schizophrenia, ASD autism spectrum disorder, ADHD attention deficit hyperactivity disorder.

genes (Fig. 3a). We identified four networks of gene-sets that consisted of both single-modality and cross-modality subsets of genes and showed significant enrichment after Bonferroni correction (Supplementary Data 10). These processes were mostly related to nervous system development and neuronal growth and differentiation. However, most gene-sets were significantly enriched of cross-modality genes only and highlight the genes' involvement in fundamental biological processes, such as cell cycle processes, cellular structure (chromatin, cytoskeleton, cell junction), and vesicle transport.

To investigate potential differential temporal patterns of cortical gene expression between single-modality and cross-modality genes, we investigated transcriptome data of post-mortem brain tissue ($n = 56$) representing males and females of multiple ethnicities across the life span (Fig. 3c)[38]. The number of fMRI-modality genes in the data ($n = 5$) was too low to generate a reliable expression pattern (Supplementary Fig. 8). Prenatal gene expression was high for both sMRI-, dMRI-modality and cross-modality genes compared to all genes in the data. In (early) childhood until adolescence, a differential dMRI-modality expression pattern was not apparent (compared to all genes in the data), whereas sMRI-modality and cross-modality genes were generally lower expressed. This matches a large body of research showing that pre- and postnatal cortical transcriptomes differ largely and pronounced prenatal expression matches the course of cortical development[39].

Given the predominant prenatal gene-expression pattern, we investigated whether single-modality and cross-modality genes were enriched for any cell-type identified by Bhaduri et al.[40] using single-cell RNA sequencing analyses of the fetal brain (Fig. 3d). We observed that cross-modality genes were enriched across all cell-types, confirming their importance in fetal brain tissue and suggesting their expression pattern is not cell-specific and more general in nature (Supplementary Data 11).

## Leveraging shared genetics with psychiatric disorders for polygenic prediction

Alterations in brain morphology, connectivity, and tissue composition often co-occur in heritable psychiatric disorders[11], suggesting that our multimodal, multivariate genetic signal may have relevance for the genetics of psychiatric disorders. It is possible to boost polygenic prediction by re-ranking the test-statistics from a given GWAS based on a genetically related secondary GWAS[41]. We therefore conditioned five major psychiatric disorder GWAS summary statistics on our multimodal MOSTest summary statistics using the conditional false discovery rate approach (condFDR; Supplementary Note 2)[42]. We included schizophrenia (SCZ)[43], bipolar disorder (BD)[44], major depressive disorder (MDD)[45,46], attention-deficit hyperactivity disorder (ADHD)[47], and autism spectrum disorder (ASD)[48] (Suppelementary Note 3; Supplementary Data 12). The rationale behind condFDR is that, in the presence of cross-trait enrichment, a variant with strong associations with both traits is more likely to represent a true association[30]. These disorder-multimodal condFDR summary statistics were used to construct polygenic scores for the disorders in independent samples using a pleiotropy-informed polygenic scoring method[30] (pleioPGS; see Methods). For that purpose, we constructed polygenic scores with PRSice2 in three independent clinical datasets (BUPGEN, TOP and MoBa, see Supplementary Data 12) based on the top 10–150,000 original disorder GWAS ordered SNPs and condFDR-based ordered SNPs (Fig. 4). For each disorder a likelihood ratio test was then applied to test if a model with both pleioPGS and original PGS provides significantly better prediction compared to the model with only one PGS included.

We observed a 1.24-fold increase in variance explained ($R^2 = 6.22\%$, $p = 2.28 \times 10^{-15}$) for BD in the TOP sample ($N_{case} = 463$; $N_{control} = 1073$) using the pleioPGS with 100,000 SNPs, compared with the original GWAS-based PGS ($R^2 = 5.02\%$, $p = 2.28 \times 10^{-15}$). The likelihood ratio test that compared the model fit before and after adding BD pleioPGS to the original BD PGS showed that pleioPGS brings a significant improvement in model fit ($\chi^2$ ($df = 2$) = 13.35, $p = 1.26 \times 10^{-3}$). Also for ADHD, the pleioPGS ($R^2 = 1.47\%$, $p = 4.57 \times 10^{-38}$) performed better than the original PGS ($R^2 = 1.20\%$, $p = 2.62 \times 10^{-31}$), contributing significantly to explaining the variation in ADHD liability ($\chi^2$ ($df = 2$) = 33.54, $p = 5.21 \times 10^{-8}$) in MoBa cases ($N = 2216$) and controls ($N = 206,644$). In

MDD cases ($N_{case} = 135$) and controls ($N_{control} = 1073$), the pleioPGS ($R^2 = 3.98\%$, $p = 2.16 \times 10^{-6}$) explained more variance than the original GWAS-based PGS ($R^2 = 3.22\%$, $p = 1.92 \times 10^{-5}$) as well. Adding MDD pleioPGS to the original MDD PGS did not improve the model fit ($\chi^2$ ($df = 2$) = 5.30, $p = 0.07$), but pleioPGS remained significant in the combined model while the original MDD PGS became insignificant (original PGS $t = 0.90$, $p = 0.37$; pleioPGS $t = 2.29$, $p = 0.02$). A similar observation was made in the case of ASD: although the higher variance explained by pleioPGS ($R^2 = 2.42\%$, $p = 3.50 \times 10^{-4}$) compared to the original PGS ($R^2 = 1.43\%$, $p = 0.01$) in BUPGEN ($N_{case} = 331$, $N_{control} = 1073$) did not cause a significantly increased model fit ($\chi^2$ ($df = 2$) = 5.93, $p = 5.17 \times 10^{-2}$), pleioPGS remained significant in the combined model while the original PGS became insignificant (original PGS $t = -0.70$, $p = 0.48$; pleioPGS $t = 2.42$, $p = 15.37 \times 10^{-3}$). When predicting SCZ in TOP ($N_{case} = 735$; $N_{control} = 1073$), the addition of the original GWAS-based PGS provided significant improvement ($\chi^2$ ($df = 2$) = 23.17, $p = 9.30 \times 10^{-6}$).

## Discussion

Our findings demonstrated that many loci and genes show pleiotropic effects across brain characteristics derived from three distinct MRI modalities. We found evidence of extensive pleiotropy across structural, functional and diffusion MRI from genetic overlap (loci and genes being associated with at least two out of three modalities), and a boost in discovery of loci ($n = 136$) and genes ($n = 384$) when all MRI-derived phenotypes are jointly analyzed using MOSTest. The results in the ABCD cohort showed generalizability of structural and diffusion MRI pleiotropic loci from adulthood to childhood, and from European ancestry to (ad)mixed ancestries. Genes with cross-modality pleiotropic effects were highly prenatally expressed, though not cell-type specific and mostly enriched in gene-sets of general biological functions. Moreover, we showed how these results can be leveraged to improve polygenic prediction of bipolar disorder and ADHD.

The human brain is a highly complex and inter-connected structure for which "the whole is more than the sum of its parts"[49]. This complexity emerges from tight interplay between different units and processes, where disturbance of any part may change the state of the whole system. With this in mind, pervasive effects of genetic variants on brain-related traits are inevitable, resulting in abundant pleiotropy not only across phenotypically linked traits such as brain morphology measures[39], but also in widespread genetic overlap between distinct aspects of brain functioning such as personality and cognition[41]. Our results implicate that the multivariate genetic signals of structural, functional, or diffusion MRI are not only composed of pleiotropic effects within modalities as previously shown[5–9], but also of a component that is shared across brain traits measured with different MRI modalities. This provides a new conceptual insight into the integration of human brain functional connectivity (fMRI), microstructure (dMRI) and macrostructure (sMRI) and highlights the importance of characterizing patterns of specificity and pleiotropy to improve our understanding of molecular neurobiological mechanisms. It is expected from theoretical analysis that rare variants would follow this pattern of pleiotropy under assumption of the infinitesimal model[50], which would be an interesting direction of future research given the recent availability of multivariate tools for gene-based rare-variant association studies[51,52].

We observed genetic overlap across all three modalities in our discovery sample from UK Biobank. Even though the heritability and number of loci and genes for fMRI was lower than in dMRI and sMRI, the proportions of genome-wide significant loci and genes that did and did not overlap with other modalities was generally equal across modalities. The results in the ABCD cohort showed generalizability of structural-diffusion MRI pleiotropy from old age to late childhood, and from European ancestry to (ad)mixed ancestries. Note that more sophisticated methods for analyses with admixed individuals are

becoming available[53], that allow to model individuals' local ancestry. A lack of power in the smaller ABCD cohort most likely limited the robust estimation of genetic associations for the relatively low heritable functional MRI-derived phenotypes, which complicated examining the generalizability of structural-functional-diffusion MRI pleiotropy beyond structural-diffusion MRI pleiotropy. The necessary future samples size to uncover the heritability of neuroimaging-derived phenotypes depends on genetic architecture and varies across modalities or the brain trait of interest[6]. For example, researchers expect that genome-wide significant variants in the prospective UK Biobank neuroimaging sample of 100,000 individuals will explain 32% and 24% of the SNP-based heritability of cortical surface area and thickness respectively[5]. This highlights that both larger neuroimaging-genetic datasets and novel statistical approaches are required to delineate the genetic background of neuroimaging phenotypes.

Our enrichment analyses showed that the biological processes and molecular components only enriched for cross-modality genes include more general functions such as cell cycle processes, regulation of gene expression, cell junctions. That gene-sets implicated by cross-modality genes alone are involved in fundamental biological processes is consistent with previous findings that genes associated with multiple trait domains are more likely to be involved in general biological functions[3]. This is in line with our finding that, although cross-modality genes were highly expressed in fetal brain tissue, those genes were not necessarily cell-specific – suggesting they may serve in cellular functions taking place in all cell-types. Future research may explore this putative relationship between the degree of pleiotropy and the specificity of biological functions. The functional enrichment of cross-modality lead SNPs investigated here can hint towards the mechanisms through which pleiotropic effects could be exerted. Cross-modality pleiotropic lead SNPs were enriched for exonic variants as found previously for pleiotropic SNPs[3,12]. There is previous evidence to suggest that pleiotropy emerges from the variants' effect on total expression of functional protein, for example by the selective exclusion of missense exons from the gene transcript[54]. However, identification of the (unmeasured) causal SNPs tagged by these multivariate GWAS lead SNPs is necessary in future studies to uncover the mechanisms through which variants exerts their pleiotropic effect.

Conditioning GWAS summary statistics for five major psychiatric disorders on our multimodal multivariate analysis prioritized variants that enhanced polygenic prediction for BD and ADHD. This highlights that future diagnostic strategies targeting psychiatric disorders may benefit from aligning the PGSs to a relevant endophenotype. This rationale is similar to a previously published study that showed improved prediction of SCZ after filtering on genetic variants expressed in the placenta[55]. Exploring which disorders benefit from either a more specific or broader range of neuroimaging phenotypes could also be an intriguing area for future research.

Some limitations are worth noting when interpreting our results. First, despite our efforts to harmonize the three sets of phenotypes to the greatest degree, the differential spatial granularity and number of features across modalities can result in differential representation of certain brain regions or brain characteristics in the multivariate signal. Second, our definition of single-modality loci and genes is inherent to two factors: 1) the grouping of phenotypes in unimodal and multimodal analyses – MOSTest may miss a small number of hits when genetic signal is sparse across the jointly analyzed phenotypes[7], and 2) the currently available data limits our statistical power – loci that are now associated with one modality could become genome-wide significant in another modality once sample sizes increase. Third, one should keep in mind that the presence of statistical pleiotropy as indicated in this study can include instances of pleiotropy where multiple traits are affected by one gene but different causal SNPs[4] or one locus with distinct gene effects that are in linkage disequilibrium[3]. Fourth, MOSTest does not provide effect directions due to its

multivariate nature and requires the use of individual level data. This restricted access to certain post GWAS analyses that require direction of effect, but these limitations were outweighed by MOSTest's ability to boost identification of variants with shared effects across phenotypes and handle hundreds of phenotypes with sample size differences in a computationally efficient manner[7]. Fifth, the temporal gene-expression patterns of single-modality and cross-modality genes are based on a postmortem dataset with a limited number of donors per timepoint[38]. A study that was recently released compared gene-expression between postmortem and living brain samples and found a significant difference in 80% of the genes[56]. Although living postnatal gene-expression samples may become available in the future for validation of our findings, prenatal gene-expression will remain to rely on postmortem brain tissue.

In conclusion, we identified extensive cross-modality pleiotropy and demonstrated that combining different neuroimaging modalities in multivariate analysis substantially increases genetic variant and gene discovery compared to multivariate analyses within single modalities. The results presented improve our understanding of the biology implicated by single-modality and cross-modality genetic effects, and provide insights into the mechanistic pathways linking common genetic variation, brain structure and function, and psychiatric disorders.

## Methods

### Samples

**UK Biobank.** The primary analyses of this study were conducted using data from UK Biobank participants who provided written informed consent. This population-based resource obtained ethical approval from the National Research Ethics Service Committee North West–Haydock (reference 11/NW/0382) and the current study was conducted under application number 27412. We included participants that passed quality control for functional[9] or diffusion[8] MRI-derived phenotypes as described in previous publications ($N_{fMRI} = 39,951$; $N_{dMRI} = 31,306$). We increased our sample size for participants with structural MRI-derived phenotypes compared to the original MOSTest publication[7], since new data had been released ($N_{sMRI} = 42,068$). For all three subsamples, we excluded participants based on relatedness (kinship coefficient >0.05 as estimated in PLINK), non-European ancestry (UKB field 22006), a genotype missing rate >10%, and bad scan quality as indicated by an (age- and sex-adjusted) Euler number >3 SDs lower than the scanner site mean. This resulted in the sample characteristics described in Supplementary Data 18.

**Adolescent brain cognitive development (ABCD) study.** Baseline data from ABCD participants from release 3.0 [NIMH Data Archive (NDA) DOI:10.151.54/ 1519007] were used for the replication efforts in this study. All children in this cohort assented before participation and their parents or guardians provided written informed consent. The procedures were approved by a central Institutional Review Board (IRB) at the University of California, San Diego, and, in some cases, by individual site Institutional Review Boards. We included participants with data for the structural or functional MRI-derived phenotypes of interest ($N_{sMRI} = 11,760$; $N_{fMRI} = 11,801$) or quality controlled diffusion MRI-derived phenotypes as previously described[8] ($N_{dMRI} = 11,904$). For all three subsamples, we excluded participants based on recommended criteria for either modality as provided by ABCD (e.g. imgincl_t1w_include), relatedness (first cousin), a genotype missing rate >10%, and bad scan quality as indicated by an (age- and sex-adjusted) Euler number <3 SDs lower than the scanner site mean. This resulted in a sample with (ad)mixed ancestries, as described by Fan et al. recently[57]. We additionally excluded participants with a genetic ancestry factor[31] of European ancestry <90% (as provided by ABCD and applied previously in Loughnan et al.[58]) to create a replication sample that matched the

ancestry characteristics of the discovery sample more specifically (Supplementary Data 18).

### Genotype data

**UK Biobank.** UK Biobank samples were genotyped from whole blood either using the UK BiLEVE or the UK Biobank axiom array and subsequently quality controlled and imputed by the UK Biobank Team[59]. Additional quality control was performed in-house and included SNP filters on minor allele frequency (MAF > 0.1%), imputation information score (INFO > 0.5), Hardy-Weinberg equilibrium (HWE; $p < 1 \times 10^{-9}$) and missingness (<10%). This resulted in 9,061,587 SNPs used for association testing. Ancestral principal components were computed within European samples by UK Biobank and used to control for population stratification.

**ABCD.** Release 3.0 genotype data from ABCD participants was obtained through the Affymetric NIDA SmokeScreen Array, using either saliva or whole blood based on higher successful calls, higher non-missingness, matched genetic sex and less excessive identity by state. Initial quality control was performed by ABCD based on calling signals and variant call rates and subsequently following pre-imputation RICOPILI (Rapid Imputation and Computational Pipeline). We complemented ABCD quality control after creating two subsamples (European and [ad]mixed ancestries as described above) by further filtering pre-imputed variants on call rates (<5% missingness), MAF > 0.01, passing the HWE test ($p < 1 \times 10^{-9}$) and heterozygosity rate (deviating >6SD from the mean value) in PLINK2[60]. A pruned set of SNPs ($r^2 = 0.1$) was used to estimate 20 genetic principal components within each subsample to use downstream as covariates in multivariate GWAS. Genetic data was phased and imputed using the TOPMed imputation server and only SNPs with high imputation quality were retained (INFO > 0.9).

### Neuroimaging data

**Structural MRI-derived phenotypes.** Three previous publications have used MOSTest on sMRI-derived phenotypes. These included either region-of-interest (ROI)-based cortical thickness, surface area, and subcortical volume[7] or vertex-based cortical thickness, surface area[5], and sulcal depth[6]. Given that the aim of this study was to combine phenotypes derived from three modalities and MOSTest can currently analyze a few thousand of phenotypes simultaneously, we opted to use the ROI-based cortical thickness, surface area, and subcortical volume[7] phenotypes given their relative low dimensionality. Supplementary Data 1 contains all the regional morphology measures included in the current study and indicates which measures were analyzed for the left and right hemisphere separately. The respective publication by Van Der Meer & Frei et al.[7] describes how the sets of 36 regional subcortical volumes, 68 cortical thickness and 68 surface area, as well as estimated intracranial volume (for covariate use downstream), were extracted from T1-weighted MRI using FreeSurfer v5.3[61,62] in UK Biobank samples. A similar procedure was performed by the ABCD Data Acquisition and Integration Core and were readily available. As the importance of normally distributed phenotypes for MOSTest was demonstrated in the original publication[7], we applied rank-based inverse-normal transformation to each measure.

**Functional MRI-derived phenotypes.** We used functional MRI (fMRI)-derived phenotypes as previously described by Roelfs et al.[9]. The UK Biobank resting-state fMRI scans were processed into 1000 Schaefer parcels[63] and mapped onto 17 large-scale brain networks defined by Yeo et al.[64]. The averaged time series within each Yeo-defined network were Pearson correlated and represented 136 brain connectivity measures next to the 17 network variances (Supplementary Data 1). Rank-based inverse-normal transformation was applied to each measure.

The ABCD Data Acquisition and Integration Core provided similar, but not identical, resting-state fMRI-derived phenotypes for replication purposes. Instead of the 17 Yeo & Krienen networks based on 1000 parcels, temporal variance in 333 Gordon-defined parcels and 66 average correlations between 12 Gordon-defined networks were available[65]. We averaged the parcel variances belonging to the same network to achieve comparability to our discovery phenotypes. Subsequently, we rank-based inverse-normal transformed the 12 network variances and 66 network connectivity phenotypes.

**Diffusion MRI-derived phenotypes.** We used diffusion MRI-derived phenotypes from UK Biobank and ABCD based on a voxel-wise restriction spectrum imaging (RSI) model (as in Fan et al.[8]). In short, RSI estimates the signal volume fractions of separable pools of water in the human brain (i.e. intracellular, extracellular, and unhindered free water) and their corresponding spherical harmonic coefficients[66,67]. Three RSI features were used: 1) restricted isotropic diffusion (N0) is most sensitive to isotropically diffusing water in the restricted compartment (within cell bodies), restricted directional diffusion (ND) is sensitive to anisotropically diffusing water in the restricted compartment (within oriented structures such as axons and dendrites), and 3) normalized free water diffusion (NF) is sensitive to cerebrospinal fluid or intravascular spaces[68]. Fan et al.[8] calculated the principal components (PCs) across all voxels and extracted the first 5000 PCs explaining more than 70% of the total variance of each feature. Here, due to dimensionality constraints, we reduced the number of PCs for each feature by estimating the "elbow" of each scree plot of eigenvalues using the nScree function of the nFactors R package (Supplementary Fig. 10). This resulted in the first 124 ND-PCs, 87 NF-PCs and 65 N0-PCs used in our multivariate GWAS.

**Statistical analyses**
**SNP-based GWAS.** We performed discovery and replication SNP-based GWAS in PLINK2[60] for every MRI-derived phenotype separately while controlling for sex, age$^2$ genotype array (UKB only), scanner, 20 genetic principal components, and modality specific covariates. The latter included Euler number, and total surface area, mean thickness or intracranial volume (sMRI), signal to noise ratio and motion (fMRI), and intracranial volume (dMRI). A linear regression model with additive allelic effects was fitted for each SNP. Subsequently, SNP-based heritability ($h^2_{SNP}$) was estimated for each phenotype using Linkage Disequilibrium Score Regression (LDSC)[32]. Univariate GWAS summary statistics from non-heritable phenotypes (nominal significance threshold $\frac{h^2_{SNP}}{h^2_{SNP\ SE}} > 1.96$ as used by Roelfs et al.[9]) were dropped from further multivariate analyses, since including them may reduce statistical power[7]. This led to the exclusion of 3 dMRI-derived NF-PCs and 14 fMRI-derived connectivity phenotypes GWAS summary statistics (Supplementary Data 1). Then, variant z-scores from univariate GWAS were combined in the MOSTest framework to construct multivariate p-values as described by Shadrin et al.[5]. This approach selects a regularization parameter optimized to the maximum yield of genetic loci (Supplementary Data 19). The alpha level for SNPs reaching genome-wide significance in the multivariate GWAS was $\alpha = (5 \times 10^{-8})/3$.

**Locus definition.** We defined genome-wide significant loci from MOSTest and conditional FDR summary statistics following a protocol as implemented in FUMA[69]. First, independent genome-wide significant SNPs ($p < 5 \times 10^{-8}/3$) were obtained by clumping ($r^2 < 0.6$) and SNPs in linkage disequilibrium (LD) with them ($r^2 \geq 0.6$) were defined as candidate SNPs. LD was estimated using reference genotypes, using 5000 random participants from the UK Biobank sample for UKB-based summary statistics and 1000 Genomes Phase 3 EUR for the European as well as (ad)mixed ancestry ABCD-based summary statistics. Second, independent significant SNPs with $r^2 < 0.1$ were defined as lead SNPs

and the minimum and maximum positional coordinates of the corresponding candidate variants defined the locus start and end position. Loci in <250 kb proximity were merged into a single locus. We excluded loci with a single SNP as these are more likely to be false positives. To test the robustness of our findings we performed sensitivity analyses for the locus definition parameters LD cut-off and merging proximity window, leading to twelve scenarios with different locus definition settings (Supplementary Fig. 3).

**Multivariate gene-based GWAS.** We explored the overlap between modalities and multimodal MOSTest on a gene-level by applying a SNP-wise mean model for 18,877 genes with MAGMA (Multi-marker Analysis of GenoMic Annotation) v1.08[33] in FUMA[69]. The SNP-based MOSTest summary statistics from sMRI, dMRI, fMRI and multimodal served as input with default settings and the UKB European population was used as reference. The alpha level for genes reaching genome-wide significance was adjusted from $\alpha = 0.05$ to $\alpha = (0.05/18,877)/3 = 8.83 \times 10^{-7}$ according to Bonferroni correction for multiple testing.

**Definition of single-modality and cross-modality loci and genes.** Locus overlap between the three MOSTest summary statistics (sMRI, fMRI, dMRI) was defined as physically overlapping genome-wide significant loci after clumping (see above). We used the GenomicRanges R-package[70] to compare the chromosome and start and end base pair positions of all loci between any pair of summary statistics. A locus was considered single-modality when it did not overlap with any of the loci identified for other modalities. All loci that were found to overlap between two or more modalities also overlapped with the multimodal loci, hence we decided to represent these cross-modality loci with the association statistics of the multimodal locus' lead SNP(s) in downstream analyses. All univariate statistics that laid the basis of these single-modality and cross-modality lead SNPs were plotted in a heatmap with the hierarchical clustering algorithm as implemented in seaborn.clustermap (https://seaborn.pydata.org/generated/seaborn.clustermap.html) applied to the phenotypes using (Pearson's) correlation as a distance metric and method = 'average' clustering. The sMRI, fMRI, dMRI and multimodal MOSTest genes that were found to be genome-wide significant in MAGMA were compared to provide a similar overview. The overlapping patterns were then plotted with the eulerr R-package. This procedure was repeated for both discovery (UKB) and replication (ABCD) MOSTest summary statistics, to investigate whether a similar overlapping pattern could be observed.

**Comparison of single-modality and cross-modality lead SNPs.** We were interested in potential differences between single-modality and cross-modality loci compared to other complex traits. Therefore we selected the lead SNPs within the respective loci (see *Locus definition*) and annotated them with ANNOVAR[34]. As reference, we used the annotations of 43,492 (unique) lead SNPs derived from 558 traits (with reasonable power, i.e. $N > 50,000$) across 24 trait domains (Table 2 from Watanabe et al.[3]). Enrichment of the single-modality and cross-modality lead SNPs in positional annotation categories was then tested using Fisher's Exact test. The alpha level for significant enrichment was Bonferroni corrected ($\alpha = 0.05/44 = 1.14 \times 10^{-3}$).

**Comparison of single-modality and cross-modality gene properties.** In order to interpret the biological processes, cellular components or molecular functions our single-modality and cross-modality genes are involved in, Gene Ontology (GO)[36,37] gene-sets were tested for enrichment of genes in these four lists using hypergeometric testing as implemented in FUMA[69]. Protein coding genes were used as background genes and Bonferroni correction was applied to adjust for multiple comparisons. The significant single-modality and cross-modality GO terms were visualized as a graph using Cytoscape[71], EnrichmentMap[72] and AutoAnnotate[73] following the Nature Protocol

by Reimand et al.[74]. Stringent pathway similarity scores (Jaccard and overlap combined coefficient = 0.6 as used in Paczkowska, Barenboim et al.[75]) were used as edges.

To visualize the temporal gene expression pattern of the sets of single-modality and cross-modality genes, we made use of gene expression data derived from brain tissue from 56 donors[38]. This dataset ranges from 5 weeks post conception to 82 years of age and we used the data as pre-processed in Kang et al.[38]. We selected the probe with the highest differential stability for each gene ($n = 16,849$). A number of single-modality ($n_{sMRI-modality} = 111$, $n_{dMRI-modality} = 83$, $n_{fMRI-modality} = 2$) and cross-modality genes ($n = 300$) were not available in the data. Given the relatively high homogeneity of expression patterns across cortical brain samples[76], we subsequently averaged over 13 cortical regions, within donor, and normalized the expression values, within probe, across donors, to a range between 0 and 100. Mean expression over time per set of genes was plotted with ggplot2 in R v4.0.3., with geom_smooth(method = "gam") using default settings.

Lastly, we explored cell-type enrichment of single-modality and cross-modality genes by performing Fisher Exact Tests for each cell-type identified by Bhaduri et al.[40]. Using single-cell RNA sequencing analyses of the fetal brain. The minimum number of genes overlapping was set to >2. Bonferroni correction was applied for the number of tests performed ($\alpha = 0.05/30 = 1.67 \times 10^{-3}$).

**Conditional FDR.** We explored whether the multimodal multivariate summary statistics could be leveraged to improve polygenic prediction for major psychiatric disorders (schizophrenia[43], bipolar disorder[44], major depressive disorder[45,46], attention-deficit hyperactivity disorder[47], and autism spectrum disorder[48]). For that purpose, we applied Conditional False Discovery Rate (cFDR)[42] on Psychiatric Genomics Consortium GWAS summary statistics (listed in Supplementary Data 12) by conditioning on multimodal MOSTest summary statistics. We obtained disorder summary statistics excluding UK Biobank to prevent sample overlap. In cFDR analyses, original $p$-values are replaced by FDR values that reflect the posterior probability that a SNP is null for the disorder given that the $p$-values for both phenotypes are as small or smaller as the observed $p$-values:

$$FDR(p_{disorder}|p_{multimodal}) = \frac{\pi_0(p_{multimodal})p_{disorder}}{F(p_{disorder}|p_{multimodal})} \quad (1)$$

with F = the conditional empirical cumulative distribution function and $\pi_0(p_{multimodal})$ = the conditional proportion of null SNPs for the disorder given that $p$-values for the multimodal phenotype are as small or smaller.

**PleioPGS.** We then computed the prediction power of the original and conditioned summary statistics by constructing polygenic scores (PGS) in independent case-control samples for the five major psychiatric disorders described above in independent samples. The TOP, BUPGEN and MoBa[77] samples are described in the Supplementary Methods (and Supplementary Data 12). We applied two different setups, both based on the C + T (clumping + thresholding) approach[78] using different strategies for ranking SNPs: 1) original GWAS $p$-value-based ranking and original GWAS effect sizes (standard PGS); 2) cFDR-based ranking (described above) and original GWAS effect sizes (pleioPGS as introduced by Van der Meer et al.[30]; https://github.com/precimed/pleiofdr). For these two setups PGS were calculated across five sets of LD-independent SNPs ($n_{SNP} = 1000$, 10,000, 50,000, 100,000, 150,000) using PRSice-2 (v2.3.3)[79] with no additional clumping (--no-clump option). Sets of LD-independent SNPs were obtained using plink v1.90b6.1[60] based on the setup-defined SNP ranking with --clump-kb 250, --clump-r2 0.1 parameters and in-sample LD estimates. In both setups the phenotypic variance explained by the PGS ($R^2$) was estimated as the difference between the $R^2$ of the full regression model and the $R^2$ of the null model only including the covariates (age, sex and the first 10 genetic PCs), also known as Nagelkerke's pseudo-$R^2$. Bonferroni correction was applied for the number of tests ($p < (0.05/(7\ n_{SNP}\ cut-offs \times 2\ PRS\ models \times 5\ disorders) =) 7.14 \times 10^{-4}$.

Per disorder we tested if the highest $R^2$ (MDD pleioPGS/SCZ original PGS with $n_{SNP} = 50,000$; ADHD/ASD/BD pleioPGS with $n_{SNP} = 100,000$) was significantly higher than the $R^2$ of the alternative PGS (MDD original PGS/SCZ pleioPGS with $n_{SNP} = 50,000$; ADHD/ASD/BD original PGS with $n_{SNP} = 100,000$). For each disorder a likelihood ratio test was applied to test if a model with both pleioPGS and original PGS provides significantly better prediction compared to the model with only one PGS included.

### Reporting summary
Further information on research design is available in the Nature Portfolio Reporting Summary linked to this article.

### Data availability
The genome-wide summary statistics generated in this study have been made publicly available via https://cncr.nl/research/summary_statistics/ and GWAS Catalog (http://ftp.ebi.ac.uk/pub/databases/gwas/summary_statistics/GCST90319001-GCST90320000/GCST90319487/, http://ftp.ebi.ac.uk/pub/databases/gwas/summary_statistics/GCST90319001-GCST90320000/GCST90319488/, http://ftp.ebi.ac.uk/pub/databases/gwas/summary_statistics/GCST90319001-GCST90320000/GCST90319489/, http://ftp.ebi.ac.uk/pub/databases/gwas/summary_statistics/GCST90319001-GCST90320000/GCST90319490/). The individual-level data that support the discovery findings of this study are available from UK Biobank but restrictions apply to the availability of these data, which were used under license no. 27412 for the current study. All researchers who wish to access this resource must register with UK Biobank by completing the registration form in the Access Management System. Data used in the preparation of this article were obtained from the Adolescent Brain Cognitive Development[SM] (ABCD) Study (https://abcdstudy.org), held in the NIMH Data Archive (NDA). ABCD data used for replication in this study is registered under the NDA study register at https://doi.org/10.15154/1527969. ABCD data is publicly shared with eligible researchers that have completed the data access application through their NDA account, further outlined here: https://wiki.abcdstudy.org/faq/faq.html. Data from the Norwegian Mother, Father and Child Cohort Study and the Medical Birth Registry of Norway used in this study are managed by the national health register holders in Norway (Norwegian Institute of public health) and can be made available to researchers, provided approval from the Regional Committees for Medical and Health Research Ethics (REC), compliance with the EU General Data Protection Regulation (GDPR) and approval from the data owners. All data generated during this study are included in this published article and its supplementary information files.

### Code availability
Code to obtain the results presented in this manuscript are available via https://github.com/EPTissink/MOSTest-multimodal[80].

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

## Acknowledgements

E.P.T. has been supported by the Foundation "De Drie Lichten" and The Simons Foundation Fund in The Netherlands. This work was partly performed on the TSD (Tjeneste for Sensitive Data) facilities, owned by the University of Oslo, operated and developed by the TSD service group at the University of Oslo, IT Department (USIT) (tsd-drift@usit.uio.no). We gratefully acknowledge support from the The Netherlands Organization for Scientific Research (NWO Gravitation: BRAINSCAPES: A Roadmap from Neurogenetics to Neurobiology - Grant No. 024.004.012, and VICI 452-16-015), The European Research Council (Advanced Grant No ERC-2018-AdG GWAS2FUNC 834057, Consolidator Grant No 101001062), American National Institutes of Health (NS057198, EB000790, 1R01MH124839, R01MH120025, R01MH122688), The Research Council of Norway (RCN) (229129, 213837, 324252, 300309, 273291, 223273, 248980, 276082, 323961), The South-East Norway Regional Health Authority (2019-108, 2022-073), KG Jebsen Stiftelsen (SKGJ-MED-021), EEA (#EEA2018-0573) collaborative grant in ASD: "Improving quality of life for Autism Spectrum Disorders patients by promoting strategies for early diagnosis and preventive measures" and EEA-RO-NO-Grant 2014–2021 (contract No 6/2019), and the European Union's Horizon 2020 research and innovation programme (No 847776 and 964874 and 801133, Marie Sklodowska-Curie grant agreement). We want to acknowledge the participants and investigators of the UK Biobank, ABCD Study, TOP, BUPGEN, PGC and FinnGen studies. The Norwegian Mother, Father and Child Cohort Study is supported by the Norwegian Ministry of Health and Care Services and the Ministry of Education and Research. We are grateful to all the participating families in Norway who take part in this on-going cohort study. We thank the Norwegian Institute of Public Health (NIPH) for generating high-quality genomic data. This research is part of the HARVEST collaboration, supported by the Research Council of Norway (#229624). Data used in the preparation of this article were obtained from the Adolescent Brain Cognitive Development SM Study (ABCD Study®) (https://abcdstudy.org), held in the NIMH Data Archive (NDA). This is a multisite, longitudinal study designed to recruit more than 10,000 children aged 9-10 and follow them over 10 years into early adulthood. The ABCD Study is supported by the National Institutes of

Health and additional federal partners under award numbers: U01DA041022, U01DA041028, U01DA041048, U01DA041089, U01DA041106, U01DA041117, U01DA041120, U01DA041134, U01DA041148, U01DA041156, U01DA041174, U24DA041123, and U24DA041147. A full list of supporters is available at https://abcdstudy.org/federal-partners/. A listing of participating sites and a complete listing of the study investigators can be found at https://abcdstudy.org/principal-investigators.html. ABCD Study consortium investigators designed and implemented the study and/or provided data but did not necessarily participate in analysis or writing of this report. This manuscript reflects the views of the authors and may not reflect the opinions or views of the NIH or ABCD Study consortium investigators. The ABCD data repository grows and changes over time. The ABCD data used in this came from [NIMH Data Archive Digital Object Identifier (10.15154/1519007)].

## Author contributions

E.P.T., A.A.S., O.F., D.v.d.M. and O.A.A. conceived of the study. E.P.T. performed the analyses with A.A.S., N.P. and D.v.d.M. Neuroimaging data were processed by D.R., T.K., C.C.F., A.M.D, D.v.d.M and E.P.T. Genotype data were processed by E.P.T. and D.v.d.M. Conceptual input on methods and/or interpretation of results was contributed by E.P.T., A.A.S., O.F., D.v.d.M., N.P., G.H., M.N., T.N., M.B., S.D., L.W., M.v.d.H., D.P., D.R., T.K., C.C.F., A.M.D, and O.A.A. E.P.T. wrote the manuscript and all authors contributed to the final manuscript.

## Competing interests

E.P.T., A.A.S., D.v.d.M, N.P., G.H., D.R., O.F., C.C.F., M.N., T.N., M.B., S.D., L.T.W., M.P.v.d.H., D.P. and T.K. declare no conflicts of interest. Dr. Andreassen has received speaker's honorarium from Lundbeck, Janssen, Otsuka and Sunovion, and is a consultant to Cortechs.ai. and Precision Health AS. Dr. Dale is a Founder of and holds equity in CorTechs Labs, Inc, and serves on its Scientific Advisory Board. He is a member of the Scientific Advisory Board of Human Longevity, Inc. and receives funding through research agreements with General Electric Healthcare and Medtronic, Inc. The terms of these arrangements have been reviewed and approved by UCSD in accordance with its conflict of interest policies.

## Additional information

[1]Department of Complex Trait Genetics, Center for Neurogenomics and Cognitive Research, Vrije Universiteit Amsterdam, Amsterdam Neuroscience, 1081 HV Amsterdam, The Netherlands. [2]Department of Sleep and Cognition, Netherlands Institute for Neuroscience, an institute of the Royal Netherlands Academy of Arts and Sciences, Amsterdam, The Netherlands. [3]NORMENT Centre, Division of Mental Health and Addiction, Oslo University Hospital and Institute of Clinical Medicine, University of Oslo, Building 48, Oslo, Norway. [4]School of Mental Health and Neuroscience, Faculty of Health, Medicine and Life Sciences, Maastricht University, Maastricht, The Netherlands. [5]Psychosis Studies, Institute of Psychiatry, Psychology and Neurosciences, King's College London, 16 De Crespigny Park, London SE5 8AB, United Kingdom. [6]Laureate Institute for Brain Research, Tulsa, OK, USA. [7]Department of Radiology, University of California San Diego, La Jolla, CA 92037, USA. [8]K.G. Jebsen Centre for Neurodevelopmental disorders, Division of Paediatric Medicine, Institute of Clinical Medicine, University of Oslo, Building 31, Oslo, Norway. [9]Prof. Dr. Alex Obregia Clinical Hospital of Psychiatry, Bucharest, Romania. [10]"Victor Babes" National Institute of Pathology, Bucharest, Romania. [11]Department of Medical Genetics, Oslo University Hospital, Oslo, Norway. [12]Department of Psychology, University of Oslo, Oslo, Norway. [13]Department of Child and Adolescent Psychology and Psychiatry, section Complex Trait Genetics, Amsterdam Neuroscience, VU University Medical Centre, Amsterdam, The Netherlands. [14]Department of Psychiatry and Psychotherapy, Tübingen Center for Mental Health, University of Tübingen, Tübingen, Germany. [15]Center for Multimodal Imaging and Genetics, University of California San Diego, La Jolla, CA 92037, USA. [16]Department of Neurosciences, University of California San Diego, La Jolla, CA 92037, USA. ✉e-mail: e.p.tissink@vu.nl; ole.andreassen@medisin.uio.no

