## [Peer Review File · Nature Communications]

Abundant pleiotropy across neuroimaging modalities identified through a multivariate genome-wide association studyReviewer #1 (Remarks to the Author):

The article by Tissink et al., investigates pleiotropy across brain MRI images or modalities (structural, diffusion, functional). The authors used MOSTest to generate GWAS summary across all imaging derived phenotypes, for each modality. Then, the authors quantified pleiotropy as the proportion of overlapping loci between the three sets of multivariate summary statistics. Additional analyses, include enrichment, gene expression, and gene based (MAGMA) analyses – with an attempt to contrast results from specific vs. pleiotropic loci. Finally, the authors applied cFDR to try to enrich/boost GWAS discovery in the context of psychiatric disorders. The out of sample evaluation, using PRS yielded mixed results.

I found the article well written, and easy to read. The amount and quality of the work should be noted. The figures, and graphical abstract are of high quality. The authors provided an appropriate level of information, which make the analyses replicable. The use of the UKB and ABCD is great, although I am not sure that the fMRI data is comparable (different atlas used).

On the downside, I have a couple methodological questions that I have raised below. More generally, I found the pleiotropy results not really ground-breaking and the analyses have been performed at a multivariate level, which implies that the results are quite “meta” and not really specific to any brain region or network. Maybe I missed some of the subtlety of the omics analyses, but I found that additional analyses did not add much to the story. I was not convinced by the results of the PRS analysis, which would have given an interesting clinical spin to the article.

Methodological questions:

- I found that your approach to multiple testing was quite lenient or not necessarily well justified. For example:

“The alpha level for SNPs reaching genome-wide significance in the multivariate GWAS was $\alpha = 5e-8$.” This does not seem to account for the 3 modalities.

“ $\alpha = (0.05/18,877) = 2.65 \times 10^{-6}$ according to Bonferroni correction for multiple testing.” Again, this does not seem to account for the 3 modalities.

“The alpha level for significant enrichment was Bonferroni corrected ($\alpha = 0.05 / 41 = 1.22 \times 10^{-3}$).” Why controlling by 41? In Figure 2 you present 13×4 tests.

“Protein coding genes were used as background genes and correction for multiple comparisons was performed using the Benjamini-Hochberg method”
Why using FDR here and not FWER as in rest of analyses?

“The number of genome-wide significant lead SNPs from UK Biobank that replicated at nominal significance ($p < 0.05$) in ABCD-based MOSTest summary statistics differed across modalities (EUR: 24.46% sMRI, 8.70% fMRI, 23.63% dMRI), and were higher for the larger sample with mixed ancestries (42.12% sMRI, 15.38% fMRI, 35.39% dMRI; Supplementary Table 6).”
You do not seem to control for multiple testing in the replication analysis.

When comparing PRS performance, how many tests did you perform? What was the significance level you used?

- The conclusion of widespread pleiotropy can be highly dependent on the loci definition. I am thinking, in particular, about the 250mbp window used, which is not based on LD (hence may consider as 1 loci several independent regions).

Have you performed sensitivity analyses (on LD cut-off and window size), or applied colocalization methods to show that the results were robust to the loci definition, or the method used?

- Pleiotropy could also be inflated by ancestry, especially that the GWAS come from the same dataset. I noted you used PLINK for the GWAS, I wonder if a LMM GWAS be compatible with MOSTest while reducing the ancestry contamination?

Could you provide qqplots of the MOSTest multivariate GWAS or other analyses/results that would suggest ancestry has been appropriately controlled for? Another idea could be to look at MAF invariance across ancestry for the significant loci?

- cFDR and PRS analysis. First I was surprised by the number of significant hits you report from the PGC results (Stable 11, 12). E.g. only 2 in MDD, while the Wray et al. paper claims 44. Note that there is a newer GWAS published in 2019 (<https://www.nature.com/articles/s41593-018-0326-7>). The reduction in number could come from excluding the UKB – but I do not think it has been used in most PGC GWAS.

For example <https://www.ncbi.nlm.nih.gov/pmc/articles/PMC9392466/#SD1> does not seem to use the UKB and it identified 287 loci.

Second, I did not find the results of the tests that evaluated to the difference in prediction between the two PRS. Thirdly, the fact that the method only works for one psychiatric disorder does not give me confidence in the overall results. Finally, you tried to validate the new loci by looking at sign concordance only, which is at best an indirect and limited measure of “replication”. What about testing their association in psychiatric cohorts?

Minor:

(Supplementary Table 16)

P4 of the supplementary does not point to the correct table

Is the ABCD mixed ancestry sample composed of a single admixture population or a mix of several populations? In the latter case, are GWAS analyses conclusive? A genetic PC plot could be informative.

The modality specific loci could be just due to a lack of power or the way IDPs are grouped – e.g. signal too region specific to show up in MOSTest. I would tone down some of the claims/discussions about these “modality specific” hits.

sign concordance tests in independent disorder summary statistics.

Sorry, I did not get why you are not relying on actual replication tests?

Stable 14 – $r^2 \gg 1$. R2 and coefficient columns may have been swapped?

The pleiotropy you reported is limited to common SNPs, and it could be that a different pattern may emerge when looking at rarer variants?

Reviewer #2 (Remarks to the Author):

Tissink et al. conducted multivariate GWAS on multiple phenotypes of different MRI modalities. They identified pleiotropic genetic associations and discussed some biological interpretations. Identified candidate genes appear to be enriched in prenatal brain development. The multimodal analysis can also assist genetic mapping in psychiatric disorders. The analysis in general is interesting, and I have the following major concerns or comments.

Regarding the results:

1. Generally, multivariate analysis is always tricky to interpret. I very much hope that the authors can try to dig further into the multivariate associations and identify the genotype-phenotype maps in a more clear manner. Where did the multivariate power come from? Which trait particularly are the SNPs affecting, especially for the cross-modality associations? It would be much better to have some illustrations of the genetic effects directions so that we understand the genetic correlations caused by the pleiotropic loci better.

2. The functional consequences of the lead SNPs using VEP seem striking to me. It appears to be common sense that most GWAS associations sit on non-coding regions, normally centered around transcription start sites of genes. Due to LD, one would expect that most of the lead SNPs would be in the non-coding regions or synonymous, and very few would be missense/non-synonymous. Thus for a GWAS analysis, I would not expect a significant enrichment of missense variants as shown in Figure 2. How about changing the background from random sets of pruned SNPs to GWAS hits of other types of human complex traits? This is asking whether the observed functional consequences distributions are special for the MRI phenotypes.

3. The lifespan expression patterns in Figure 3b look interesting. Is there any other way to validate the cross-modality genes identified in terms of this perspective? To what extent can we justify their roles in fundamental biological processes and prenatal brain development?

4. Regarding the shared genetic architecture with psychiatric disorders, the authors focused on genetic discovery conditioned on the MRI multivariate associations. The idea is fine, but I don't think the results have been shown clearly. I expect e.g., an additional main figure showing 1) the boost of discovery power, 2) the new discoveries conditioned on loci from different MRI modalities; 3) what have we missed in the standard GWAS for psychiatric disorders, including heritability and biological mechanisms?

5. Following the above, how do the findings of this study relate to or build upon previous research on genetic pleiotropy in the context of neuroimaging and psychiatric disorders? Are there any inconsistencies or discrepancies that warrant further investigation?

6. Are there any potential limitations or biases in the gene expression data used to determine the temporal and spatial specificity of the cross-modality genes, and how might these limitations impact the interpretation of the results?

Regarding the methods:

7. Regarding the multivariate test method, how well does the Multivariate Omnibus Statistical Test (MOSTest) account for potential confounding factors, such as population stratification? Genomic PCs were included as covariates in the univariate analysis, but I'm not sure whether that is sufficient for the multivariate analysis.

8. Given the sample sizes of the UK Biobank and ABCD study, can the authors address the potential impact of statistical power on the robustness of their findings, and perhaps discuss the sample size required in the future to better reveal the genetic basis of MRI phenotypes?

Regarding the discussions:

9. Can the authors discuss the implications of their findings for the development of future diagnostic and therapeutic strategies targeting psychiatric disorders, and highlight potential areas for further research?

Please find below our point-by-point response to the comments raised by the reviewers together with the adjustments we made to the manuscript. The questions and comments of the reviewer are presented in italics, our response in normal typeset. We marked changes to the text in blue in the updated version of our manuscript.

Reviewer #1

The article by Tissink et al., investigates pleiotropy across brain MRI images or modalities (structural, diffusion, functional). The authors used MOSTest to generate GWAS summary across all imaging derived phenotypes, for each modality. Then, the authors quantified pleiotropy as the proportion of overlapping loci between the three sets of multivariate summary statistics. Additional analyses, include enrichment, gene expression, and gene based (MAGMA) analyses – with an attempt to contrast results from specific vs. pleiotropic loci. Finally, the authors applied cFDR to try to enrich/boost GWAS discovery in the context of psychiatric disorders. The out of sample evaluation, using PRS yielded mixed results.

I found the article well written, and easy to read. The amount and quality of the work should be noted. The figures, and graphical abstract are of high quality. The authors provided an appropriate level of information, which make the analyses replicable. The use of the UKB and ABCD is great, although I am not sure that the fMRI data is comparable (different atlas used).

On the downside, I have a couple methodological questions that I have raised below. More generally, I found the pleiotropy results not really ground-breaking and the analyses have been performed at a multivariate level, which implies that the results are quite “meta” and not really specific to any brain region or network. Maybe I missed some of the subtlety of the omics analyses, but I found that additional analyses did not add much to the story. I was not convinced by the results of the PRS analysis, which would have given an interesting clinical spin to the article.

Response: We thank the Reviewer for the positive comments, and we have tried to clarify and highlight innovative aspects of the study in the revised version. We have addressed the specific comments below.

Methodological questions:

1. I found that your approach to multiple testing was quite lenient or not necessarily well justified. For example:

1.a “The alpha level for SNPs reaching genome-wide significance in the multivariate GWAS was $\alpha = 5e-8$.” This does not seem to account for the 3 modalities.

1.b “ $\alpha = (0.05/18,877 =) 2.65 \times 10^{-6}$ according to Bonferroni correction for multiple testing.” Again, this does not seem to account for the 3 modalities.

We agree with the Reviewer and have changed the alpha level to correct for the 3 modalities in our SNP and gene discovery analyses. The number of genome-wide significant loci therefore decreased from 640 to 590 (sMRI), 44 to 42 (fMRI), 562 to 512 (dMRI), and 851 to 794 (multimodal). The number of genome-wide significant genes in MAGMA therefore decreased from 1,809 to 1,620 (sMRI), 45 to 39 (fMRI), 1,638 to 1,453 (dMRI), and 2,515 to 2,242 (multimodal). These new numbers also affected our downstream analyses, and we have therefore rerun all follow-up analyses that were based on lists of significant genes or lead SNPs. These updated results reveal patterns very similar to the original analyses and do not change the conclusions of our manuscript. To ensure that our results remain

comparable to studies that adhere to the standard protocol (as we do in Supplementary Results 1), we have included the number of loci for each modality at $p < 5 \times 10^{-8}$ in Supplementary Results 1.

Results, page 6, line 157

Figure 1. Overlap of genome-wide significant a) loci and b) genes observed across neuroimaging modalities in **single-modality** and joint multimodal analyses ($p < 5 \times 10^{-8}/3$).

Results section, page 5, line 135

These unimodal multivariate analyses identified 590, 42 and 512 genome-wide significant loci associated with sMRI, fMRI and dMRI respectively ($p < 5 \times 10^{-8}/3$; Supplementary Data 2).

Results section, page 5, line 143

Applying MAGMA¹ gene-level analyses to unimodal multivariate summary statistics identified 1,620, 39 and 1,453 genome-wide significant genes ($p < 8.83 \times 10^{-7}$) for sMRI, fMRI and dMRI respectively (Supplementary Data 4). When overlapping the identified loci ($p < 5 \times 10^{-8}/3$) and genes ($p < 8.83 \times 10^{-7}$) from each modality (Methods), we observed 326 loci (36.18% of total) and 1,021 genes (41.79% of total) associated with at least two out of three modalities (Fig. 1a).

Results section, page 7, line 173

We therefore applied MOSTest across neuroimaging modalities, combining 583 phenotypes, identifying 794 genetic loci (Supplementary Data 2). The LD Score Regression intercept (1.04, SE=0.04) indicated that the inflation in these multimodal MOSTest p -values ($\lambda_{GC} = 2.83$) was driven by polygenicity and not population stratification or cryptic relatedness. Replication rates at nominal significance were comparable to previous MOSTest studies: EUR 26.45%, 37.55% (ad)mixed ancestries (see Supplementary Data 3 for rates after correction for the number of lead SNPs). One-hundred-thirty-six (15.09%) of these loci did not overlap with any of the loci identified in the unimodal multivariate analyses, suggesting that MOSTest leveraged the shared genetic signal across imaging modalities to boost the discovery of pleiotropic loci. Gene-based GWAS from MAGMA showed that of the 2,242 genome-wide significant multimodal genes, 384 (15.72%) were not discovered for unimodal gene-based GWAS (Supplementary Data 4).

1.c “The alpha level for significant enrichment was Bonferroni corrected ($\alpha = 0.05 / 41 = 1.22 \times 10^{-3}$).”
 Why controlling by 41? In Figure 2 you present 13*4 tests.

We agree that the number of tests included in this figure was unclear. In the original Figure 2 (Figure 3b in the current revised version), all single-modality and the cross-modality lead SNPs were annotated into one of VEP’s functional categories. Because some modalities yielded a low number of significant loci (i.e. few fMRI-modality lead SNPs), some VEP categories include zero fMRI-modality lead SNPs. The same is true for small VEP categories (e.g. splice sites), that also had a count of zero lead SNPs for most modalities. In these cases, we did not test for enrichment and Figure 2 was left blank. This resulted in the successful testing for enrichment in 41 instances, which lead us to $\alpha = 0.05 / 41 = 1.22 \times 10^{-3}$.

However, we realize Fisher Exact Test allows for contingency tables that include counts of zero. We have therefore rerun this analysis with the new lists of lead SNPs (due to the slight change in the threshold of genome-wide significant loci during the revision comment 1.a). Now including annotation categories that include zero lead SNPs, the alpha level changed to $\alpha = 0.05 / (11 \times 4) = 5.68 \times 10^{-4}$. We have revised the legend text of this Figure to clarify this issue. Please note that we have also changed the reference SNPs used, and annotation method (ANNOVAR) due to another comment during revision.

Results, page 9, line 239

Figure 2. b) Functional consequences of single-modality and cross-modality lead SNPs as annotated with ANNOVAR². When the null hypothesis (OR=1) could be rejected after Bonferroni correction ($p < 1.14 \times 10^{-3}$), the solid line indicates significant enrichment. Annotations of 43,492 (unique) lead SNPs derived from 558 traits across 24 trait domains from Watanabe et al. were used as reference for Fisher Exact Test (Supplementary Data 9). ncRNA = non-coding RNA.

1.d “Protein coding genes were used as background genes and correction for multiple comparisons was performed using the Benjamini-Hochberg method”. Why using FDR here and not FWER as in rest of analyses?

We have used FUMA’s default settings, which applies FDR correction, with the rationale that Bonferroni would be too conservative given the dependency of the multiple tests (gene-sets tested separately may overlap in their included genes). However, we agree with the reviewer that choosing Bonferroni multiple testing correction has been the standard throughout the manuscript and we have therefore changed our correction from FDR to Bonferroni in the current version of the manuscript. We found a decreased number of significant gene-sets (27 sMRI-modality, 4 dMRI-modality, and 44 cross-modality), because of this more stringent correction on top of the more stringent correction for MAGMA gene-based analyses as described above. We have updated the graphical representation in Figure 2a with these new results.

Results, page 9, line 240

Figure 2. a) Gene-ontology biological processes, molecular functions and cellular components that were (Bonferroni-corrected) significantly enriched for cross-modality, sMRI-modality and/or dMRI-modality genes (none of the GO terms tested showed enrichment for the 6 fMRI-modality genes). Node size reflects gene-set size, edges reflect pathway similarity scores (Methods).

Results, page 8, line 216

Next, we tested whether any difference in results from gene-set enrichment analyses with Gene Ontology (GO) biological processes, cellular components and molecular functions could be observed using the 599 sMRI-, 7 fMRI-, 432 dMRI-modality, or 1,405 cross-modality genes (Fig. 3a). We identified four networks of gene-sets that consisted of both single-modality and cross-modality subsets of genes and showed significant enrichment after Bonferroni correction (Supplementary Data 10).

These processes were mostly related to nervous system development and neuronal growth and differentiation. However, most gene-sets were significantly enriched of cross-modality genes only and highlight the genes' involvement in fundamental biological processes, such as cell cycle processes, cellular structure (chromatin, cytoskeleton, cell junction), and vesicle transport.

I.e. "The number of genome-wide significant lead SNPs from UK Biobank that replicated at nominal significance ($p < 0.05$) in ABCD-based MOSTest summary statistics differed across modalities (EUR: 24.46% sMRI, 8.70% fMRI, 23.63% dMRI), and were higher for the larger sample with mixed ancestries (42.12% sMRI, 15.38% fMRI, 35.39% dMRI; Supplementary Data 6)." You do not seem to control for multiple testing in the replication analysis.

We have indeed reported replication rates at nominal significance. The reason to do so was to allow for direct comparison with previous MOSTest papers that used this threshold, and we indeed observed similar replication rates. We agree with the reviewer that controlling for multiple testing would present a complete picture. We have therefore added a multiple testing correction for the number of loci tested for replication within each modality ($\alpha = 0.05 / n$ lead SNPs) to Supplementary Data 3.

Results, page 5, line 138

The number of genome-wide significant lead SNPs from UK Biobank that replicated at $p < 0.05$ in ABCD-based MOSTest summary statistics differed across modalities (EUR: 24.46% sMRI, 8.70% fMRI, 23.63% dMRI), and were higher for the larger sample with (ad)mixed ancestries (42.12% sMRI, 15.38% fMRI, 35.39% dMRI). Replication rates after correcting for the number of lead SNPs tested are provided in Supplementary Data 3.

Results, page 7, line 178

Replication rates at nominal significance were comparable to previous MOSTest studies: EUR 26.45%, 37.55% (ad)mixed ancestries (see Supplementary Data 3 for rates after correction for the number of lead SNPs).

I.f When comparing PRS performance, how many tests did you perform? What was the significance level you used?

We understand that it was not clear for the Reviewer how PRS performance was compared, because no formal statistical test had been applied to test if the difference in R^2 was significant. We have performed new analyses in the current version of the manuscript to address this issue. For every disorder, we selected the PRS with the highest R^2 (and $p < (0.05 / (7 n_{\text{SNP}} \text{ cut-offs} \times 2 \text{ PRS models} \times 5 \text{ disorders})) = 7.14 \times 10^{-4}$) and compared it to the corresponding PRS alternative (same n_{SNP} cut-off, same disorder, different PRS model). To statistically test this difference, we had to obtain confidence intervals around the R^2 estimates, which we obtained using 200 iterations of random resampling without replacement. This allowed us to compare the R^2 estimates with one T-test per disorder, resulting in an alpha level of $0.05/5$ disorders = 0.01. This procedure is now added to the Methods, Results, a main and a Supplementary Figure.

Methods, page 22, line 648

In both setups the phenotypic variance explained by the PGS (R^2) was estimated as the difference between the R^2 of the full regression model and the R^2 of the null model only including the covariates (age, sex and the first 10 genetic PCs), also known as Nagelkerke's pseudo- R^2 . Bonferroni correction was applied for the number of tests ($p < (0.05 / (7 \text{ n}_{\text{SNP}} \text{ cut-offs} \times 2 \text{ PRS models} \times 5 \text{ disorders})) = 7.14 \times 10^{-4}$).

We applied random sampling without replacement procedure and t -test per disorder to test if the highest R^2 (MDD pleioPGS/SCZ original PGS with $n_{\text{SNP}} = 50,000$ SNPs; ADHD/ASD/BD pleioPGS with $n_{\text{SNP}} = 100,000$ SNPs) significantly higher than the R^2 of the alternative PGS (MDD original PGS/SCZ pleioPGS with $n_{\text{SNP}} = 50,000$ SNPs; ADHD/ASD/BD original PGS with $n_{\text{SNP}} = 100,000$ SNPs). For each trait and type of PGS (original PGS, pleioPGS), we performed 200 iterations randomly sampling 2/3 of controls and 2/3 of corresponding disorder cases with replacement and applying PRSice2 to estimate R^2 . Obtained distributions of R^2 values were used to assess confidence intervals.

Results, page 12, line 297

We followed-up by testing whether our disorder-multimodal condFDR summary statistics could improve polygenic prediction of the disorders in independent samples using a pleiotropy-informed polygenic scoring method³ (pleioPGS; see Methods). For that purpose, we constructed polygenic scores with PRSice2 in two independent clinical datasets (TOP and MoBa, see Supplementary Data 12) based on the top 10-150,000 original disorder GWAS ordered SNPs and condFDR-based ordered SNPs (Fig. 3b). We observed a 1.24-fold increase in variance explained ($R^2 = 6.22\%$, $p = 2.28 \times 10^{-15}$) for BD in the TOP sample ($N_{\text{case}} = 463$; $N_{\text{control}} = 1,073$) using the pleioPGS with 100,000 SNPs, compared with the original GWAS-based PGS ($R^2 = 5.02\%$, $p = 2.28 \times 10^{-15}$). We employed a random sampling procedure to show that this was a significant improvement ($t = 42.16$, $p = 3.64 \times 10^{-101}$; Supplementary Fig. 9). Also for MDD ($N_{\text{case}} = 135$, $N_{\text{control}} = 1,073$), the pleioPGS ($R^2 = 3.98\%$, $p = 2.16 \times 10^{-6}$) significantly outperformed ($t = 14.57$, $p = 7.43 \times 10^{-33}$) the original GWAS-based PGS ($R^2 = 3.22\%$, $p = 1.92 \times 10^{-5}$). The phenotypic variance explained by PGS in ASD and ADHD was low ($R^2 < 2.43\%$) when using either the original PGS or pleioPGS (Supplementary Data 17), though a significant improvement of the pleioPGS over the original PGS could be observed for both ASD ($\Delta R^2 = 0.99\%$, $t = 30.99$, $p = 4.16 \times 10^{-78}$) and ADHD ($\Delta R^2 = 0.28\%$, $t = 46.44$, $p = 9.68 \times 10^{-109}$). When predicting SCZ in TOP ($N_{\text{case}} = 735$; $N_{\text{control}} = 1,073$), the original GWAS-based PGS explained significantly ($p = 1.44 \times 10^{-34}$) more variance in SCZ liability.

Results, page 11, line 285

Figure 3. b) Phenotypic variance explained for five psychiatric disorders explained by polygenic scores based on original disorder GWAS summary statistics (original PGS) and conditional disorder-multimodal GWAS summary statistics (pleioPGS). Note that PGS using top 10 and top 100 SNPs had too low R^2 values to be visible and are therefore not plotted, but can be found in Supplementary Data 17. Independent target samples were used for prediction (see Methods). A random sampling procedure and t -test were applied per disorder to test if the observed difference in R^2 was significant (Supplementary Fig. 9).

Supplement, page 14, line 189

Supplementary Figure 9. Boxplots of R^2 values after polygenic score (PGS) resampling analyses for schizophrenia (SCZ), bipolar disorder (BIP), depression (DEP), autism spectrum disorder (ASD) and attention-deficit hyperactivity disorder (ADHD). For each trait and type of PGS (pleioPGS in red, the original PGS in grey), 200 iterations of PRSice-2 were ran with random sampling of 2/3 controls ($N_{TOP}=715$ or $N_{MoBa}=35,214$) and 2/3 cases ($N_{SCZ}=490$; $N_{BIP}=309$; $N_{DEP}=90$; $N_{ADHD}=1,477$; $N_{ASD}=220$)

to estimate the R^2 values for both PGSs, keeping all parameters and covariates equal to our main analyses. taking for each sample 2/3 of controls and 2/3 of cases.

2. The conclusion of widespread pleiotropy can be highly dependent on the loci definition. I am thinking, in particular, about the 250mbp window used, which is not based on LD (hence may consider as 1 loci several independent regions). Have you performed sensitivity analyses (on LD cut-off and window size), or applied colocalization methods to show that the results were robust to the loci definition, or the method used?

We thank the Reviewer for the suggestion to test the robustness of our pleiotropic findings by performing sensitivity analyses for locus definition. Currently, we have used an LD cut-off of $r^2 = 0.6$ and a window size of 250 kilobases. This definition is widely used in the field and implemented in FUMA default protocol. We have now performed sensitivity analyses varying these two parameters, leading to 12 scenarios with different locus definition settings. Looking at the resulting Venn diagrams, these parameters have minute effects on the number of loci overlapping between modalities and support the robustness of the conclusion of widespread pleiotropy. We have added the results as Supplementary Figure 3.

Supplementary Figure 3. Results of sensitivity analyses with changing locus definition parameters (r^2 and window size), leading to 12 scenarios with different LD cut-offs and window-sizes.

3. Pleiotropy could also be inflated by ancestry, especially that the GWAS come from the same dataset. I noted you used PLINK for the GWAS, I wonder if a LMM GWAS be compatible with MOSTest while reducing the ancestry contamination?

We agree with the reviewer that a multivariate analysis can exacerbate confounding introduced through population stratification if it is only partly controlled by PCs in the univariate GWAS analysis. We try to minimize potential inflation due to population stratification by limiting our UKB-based discovery analysis to genetically homogeneous white British population (individuals who self-identified as white British and have very similar genetic ancestry based on a genetic PC analysis) and additionally by accounting for the first 20 genetic principal components. All univariate GWAS performed (before MOSTest was applied) had an LD Score Regression intercept below 1.03, suggesting no substantial influence of population stratification or cryptic relatedness, which is further supported by the biologically meaningful pathway analysis results and consistent findings in the independent ABCD cohort. Lastly, this study follows GWAS analytical pipeline which was developed and validated in the previously published studies focused on a single imaging modality (sMRI: van der Meer et al. (2020), dMRI: Fan et al. (2022), fMRI: Roelfs et al. (2022)).

The reviewer correctly noted that MOSTest's multivariate statistics were based on PLINK univariate statistics. The suggestion to alternatively use LMM for the univariate statistics, to allow for an additional random effect term to increase control for stratification, is unfortunately not compatible with MOSTest. MOSTest uses univariate GWAS z -statistics from original and randomly permuted genotypes to empirically access the distribution of the tests statistics under null hypothesis (see van der Meer et al., 2020). The permutation step is an essential part of MOSTest, however, available implementation of this step, is not compatible with LMM. We address the reviewer's concern about ancestry inflation in the response below.

Could you provide qqplots of the MOSTest multivariate GWAS or other analyses/results that would suggest ancestry has been appropriately controlled for? Another idea could be to look at MAF invariance across ancestry for the significant loci?

We thank the reviewer for the suggestions provided to show whether ancestry has been appropriately controlled for. We have taken this opportunity to perform a new analysis that allowed us to inspect whether any ancestry clusters would appear from a principal component analysis on our multimodal MOSTest lead SNPs. Genetic principal components were calculated for the discovery samples based on lead SNPs identified in the multimodal MOSTest analysis. Looking at the first four principal components (the figure below), we can conclude that no obvious population structure is revealed suggesting that the genetic variance in the SNPs that were found to be associated with multimodal neuroimaging-derived phenotypes were not driven by population stratification. Moreover, we ran LD Score Regression and concluded that the intercept of 1.04 (0.04) indicates that the inflation in multimodal MOSTest p -values ($\lambda_{GC} = 2.83$) was driven by polygenicity and not population stratification or cryptic relatedness.

Results, page 7, line 169

We next investigated whether combining all sMRI, fMRI, and dMRI-derived measures in one multivariate analysis generated greater statistical power to identify novel pleiotropic loci and genes which show sub-threshold associations in each unimodal multivariate analysis (Manhattan plot in Supplementary Fig. 4). We therefore applied MOSTest across neuroimaging modalities, combining 583 phenotypes, identifying 794 genetic loci (Supplementary Data 2). The LD Score Regression intercept (1.04, SE=0.04) indicated that the inflation in these multimodal MOSTest p -values ($\lambda_{GC} = 2.83$) was driven by polygenicity and not population stratification or cryptic relatedness.

4. cFDR and PRS analysis. First I was surprised by the number of significant hits you report from the PGC results (Stable 11, 12). E.g. only 2 in MDD, while the Wray et al. paper claims 44. Note that there is a newer GWAS published in 2019 (<https://www.nature.com/articles/s41593-018-0326-7>). The reduction in number could come from excluding the UKB – but I do not think it has been used in most PGC GWAS. For example <https://www.ncbi.nlm.nih.gov/pmc/articles/PMC9392466/#SD1> does not seem to use the UKB and it identified 287 loci.

The reviewer is correct that there was a discrepancy between the number of hits we report in comparison to the original PGC articles. We have improved our pipeline and hope to clarify the issue here.

- In the ADHD article, 12 genome-wide significant loci are reported: we report 12 too.
- The bipolar disorder paper reports 64 loci at $N_{\text{eff}} = 101,962$, we report 49: this is a reasonable decrease considering the exclusion of the UKB and TOP sample ($N_{\text{eff}} = 93,662$).
- The reviewer is correct that we did not use the largest possible GWAS for depression. We have now corrected this, using Levey et al. summary statistics. These summary statistics are a meta-analysis of the Million Veteran Program, PGC (Wray et al.), 23andMe, UKB and FinnGen. However,
 - the 23andMe summary statistics are not publicly available and were excluded
 - FinnGen summary statistics were already used for our condFDR replication phase
 - we could not allow sample overlap with our UKB-based MOSTest summary statistics.
 Exclusion of 23andMe, FinnGen and UKB samples decreased the sample size by 66% and therefore also the number of loci from the reported in the original publication ($n_{\text{loci}} = 178$) to $n_{\text{loci}}=20$.
- In the publicly available autism spectrum disorder summary statistics 2 genome-wide significant loci are reported (chr20 and chr8; $N_{\text{case}} = 18,000$; $N_{\text{control}} = 27,000$). In the accompanying article, the additional loci reported are discovered in a combined analysis with a follow-up sample ($N_{\text{case}} = 2,000$; $N_{\text{control}} = 142,000$) that is not publicly available.

- We have used the schizophrenia GWAS from April 2022 that reports 287 loci as the reviewer noted. We report 162 loci, which is due to several circumstances:
 - The 287 loci are based on the primary trans-ancestry GWAS ($N_{\text{case}} = 74,000$; $N_{\text{control}} = 101,000$) and uses locus definition which is different from what is used in our study. Applying our definition of the locus with European reference panel (1000Genomes), we identify 257 loci in these summary statistics.
 - Following the requirements of the condFDR method we also limit our analysis to European individuals resulting in further reduction of the sample and identified loci ($N_{\text{case}} = 53,000$; $N_{\text{control}} = 77,000$; $n_{\text{loci}} = 177$)
 - additionally we exclude the TOP sample to prevent sample overlap in the PRS analysis ($N_{\text{case}} = 51,000$; $N_{\text{control}} = 68,000$; $n_{\text{loci}} = 162$).

We have added a statement of these details in the (Supplementary) Results and have rerun the analysis where needed because of updated summary statistics.

Results, page 10, line 265

Compared to the number of genome-wide significant loci identified in the original GWAS (Fig. 3a), we observed a 5-19 fold increase in locus yield in the condFDR summary statistics (Supplementary Data 13). Note that the locus yield from the original GWAS may be lower than the number presented in the original articles, since we relied on publicly available data (e.g. excluding 23andMe) and exclude some additional cohorts to avoid sample overlap (e.g. UKB, see Supplementary Results 2).

Supplement, page 14, line 126

Summary statistics used for conditional FDR

In the original schizophrenia article, the authors report 287 genome-wide significant loci based on the primary trans-ancestry GWAS ($N_{\text{case}}=74,000$; $N_{\text{control}}=101,000$). Since our locus definition procedure is slightly different, and uses a European reference panel (1000Genomes), we report 257 loci in these summary statistics. Additionally following requirements of the condFDR method we had to limit our analyses to the European subsample ($N_{\text{case}}=53,000$; $N_{\text{control}}=77,000$; $n_{\text{loci}}=177$) and exclude TOP sample to prevent sample overlap ($N_{\text{case}}=51,000$; $N_{\text{control}}=68,000$; $n_{\text{loci}}=162$).

We also used summary statistics for major depression disorder published by Levey et al., though with exclusion of several samples. The original summary statistics are a meta-analysis of the Million Veteran Program, PGC (Wray et al.), 23andMe, UKB and FinnGen. However, the 23andMe summary statistics are not publicly available, we already use FinnGen summary statistics for our condFDR replication phase and could not allow sample overlap with our UKB-based MOSTest summary statistics. This decreased the sample size by 66% and therefore also the number of loci from the reported in the original publication ($n_{\text{loci}}=178$) to $n_{\text{loci}}=20$.

In the publicly available autism spectrum disorder summary statistics 2 genome-wide significant loci are reported (chr20 and chr8; $N_{\text{case}}=18,000$; $N_{\text{control}}=27,000$). In the accompanying article, the additional loci reported are discovered in a combined analysis with a follow-up sample ($N_{\text{case}}=2,000$; $N_{\text{control}}=142,000$) that is not publicly available.

The loss of 5 loci (64 in the original article at $N_{\text{eff}}=101,962$, we report 49 at $N_{\text{eff}}=93,662$) for bipolar disorder is due to excluding the UKB and TOP sample to prevent sample overlap. There was no sample overlap with the samples used for the original attention deficit hyperactivity disorder GWAS, and we report the same number of loci as the original article ($n_{\text{loci}}=12$).

Second, I did not find the results of the tests that evaluated to the difference in prediction between the two PRS.

No formal statistical test had been applied to test if the difference in R^2 was significant in the previous version of our manuscript. We have performed new analyses in the current version of the manuscript to address this issue. For every disorder, we selected the PRS with the highest R^2 (and $p < (0.05 / (7 n_{\text{SNP}} \text{ cut-offs} \times 2 \text{ PRS models} \times 5 \text{ disorders})) = 7.14 \times 10^{-4}$) and compared it to the corresponding PRS alternative (same n_{SNP} cut-off, same disorder, different PRS model). To statistically test this difference, we had to obtain confidence intervals around the R^2 estimates, which we obtained using 200 iterations of random resampling of individuals without replacement. This allowed us to compare the R^2 estimates with one T-test per disorder, resulting in an alpha level of $0.05/5 \text{ disorders} = 0.01$. This procedure is now added to the Methods, Results, a main and a supplementary Figure.

Methods, page 22, line 648

In both setups the phenotypic variance explained by the PGS (R^2) was estimated as the difference between the R^2 of the full regression model and the R^2 of the null model only including the covariates (age, sex and the first 10 genetic PCs), also known as Nagelkerke's pseudo- R^2 . Bonferroni correction was applied for the number of tests ($p < (0.05 / (7 n_{\text{SNP}} \text{ cut-offs} \times 2 \text{ PRS models} \times 5 \text{ disorders})) = 7.14 \times 10^{-4}$).

We applied random sampling without replacement procedure and t -test per disorder to test if the highest R^2 (MDD pleioPGS/SCZ original PGS with $n_{\text{SNP}} = 50,000$ SNPs; ADHD/ASD/BD pleioPGS with $n_{\text{SNP}} = 100,000$ SNPs) significantly higher than the R^2 of the alternative PGS (MDD original PGS/SCZ pleioPGS with $n_{\text{SNP}} = 50,000$ SNPs; ADHD/ASD/BD original PGS with $n_{\text{SNP}} = 100,000$ SNPs). For each trait and type of PGS (original PGS, pleioPGS), we performed 200 iterations randomly sampling 2/3 of controls and 2/3 of corresponding disorder cases with replacement and applying PRSice2 to estimate R^2 . Obtained distributions of R^2 values were used to assess confidence intervals.

Results, page 12, line 295

We followed-up by testing whether our disorder-multimodal condFDR summary statistics could improve polygenic prediction of the disorders in independent samples using a pleiotropy-informed polygenic scoring method³ (pleioPGS; see Methods). For that purpose, we constructed polygenic scores with PRSice2 in two independent clinical datasets (TOP and MoBa, see Supplementary Data 12) based on the top 10-150,000 original disorder GWAS ordered SNPs and condFDR-based ordered SNPs (Fig. 3b). We observed a 1.24 fold increase in variance explained ($R^2 = 6.22\%$, $p = 2.28 \times 10^{-15}$) for BD in the TOP sample ($N_{\text{case}} = 463$; $N_{\text{control}} = 1,073$) using the pleioPGS with 100,000 SNPs, compared with the original GWAS-based PGS ($R^2 = 5.02\%$, $p = 2.28 \times 10^{-15}$). We employed a random sampling procedure to show that this was a significant improvement ($t = 42.16$, $p = 3.64 \times 10^{-101}$; Supplementary Fig. 9). Also for MDD ($N_{\text{case}} = 135$, $N_{\text{control}} = 1,073$), the pleioPGS ($R^2 = 3.98\%$, $p = 2.16 \times 10^{-6}$) significantly outperformed ($t = 14.57$, $p = 7.43 \times 10^{-33}$) the original GWAS-based PGS ($R^2 = 3.22\%$, $p = 1.92 \times 10^{-5}$). The phenotypic variance explained by PGS in ASD and ADHD was low ($R^2 < 2.43\%$) when using either the original PGS or pleioPGS (Supplementary Data 17), though a significant improvement of the pleioPGS over the original PGS could be observed for both ASD ($\Delta R^2 = 0.99\%$, $t = 30.99$, $p = 4.16 \times 10^{-78}$) and ADHD ($\Delta R^2 = 0.28\%$, $t = 46.44$, $p = 9.68 \times 10^{-109}$). When predicting SCZ in TOP ($N_{\text{case}} = 735$; $N_{\text{control}} = 1,073$), the original GWAS-based PGS explained significantly ($p = 1.44 \times 10^{-34}$) more variance in SCZ liability.

Figure 3. b) Phenotypic variance explained for five psychiatric disorders explained by polygenic scores based on original disorder GWAS summary statistics (original PGS) and conditional disorder-multimodal GWAS summary statistics (pleioPGS). Note that PGS using top 10 and top 100 SNPs had too low R^2 values to be visible and are therefore not plotted, but can be found in Supplementary Data 17. Independent target samples were used for prediction (see Methods). A random sampling procedure and t -test were applied per disorder to test if the observed difference in R^2 was significant (Supplementary Figure 9).

Supplementary Figure 9. Boxplots of R^2 values after polygenic score (PGS) resampling analyses for schizophrenia (SCZ), bipolar disorder (BIP), depression (DEP), autism spectrum disorder (ASD) and attention-deficit hyperactivity disorder (ADHD). For each trait and type of PGS (pleioPGS in red, the

original PGS in grey), 200 iterations of PRSice were ran with random sampling of 2/3 controls ($N_{TOP}=715$ or $N_{MoBa}=35,214$) and 2/3 cases ($N_{SCZ}=490$; $N_{BIP}=309$; $N_{DEP}=90$; $N_{ADHD}=1,477$; $N_{ASD}=220$) to estimate the R^2 values for both PGSs, keeping all parameters and covariates equal to our main analyses. The statistical difference between the mean was tested using t-tests (see Main Results).

Thirdly, the fact that the method only works for one psychiatric disorder does not give me confidence in the overall results.

Inspired by the previous two comments of the reviewer, we adapted the use of most up-to-date summary statistics and improved our pipeline to test the improvement of pleioPGS over the original PGS. With these two changes in place, we are happy to observe that pleioPGS statistically outperforms the original PGS in four out of five psychiatric disorders.

Discussion, page 14, line 375

When conditioning GWAS summary statistics for five major psychiatric disorders on our multimodal multivariate analysis, we discovered new loci including genes with increased brain-specific gene expression. The validity of these boosted loci needs to be replicated in independent samples in future research. Nevertheless, prioritizing these variants already showed enhanced polygenic prediction for MDD, BD, ASD and ADHD. This highlights that future diagnostic strategies targeting psychiatric disorders may benefit from aligning the PGSs to a relevant endophenotype. This rationale is similar to a previously published study that showed improved prediction of SCZ after filtering on genetic variants expressed in the placenta⁴. Exploring which disorders benefit from either a more specific or broader range of neuroimaging phenotypes, could also be an intriguing area for future research.

Finally, you tried to validate the new loci by looking at sign concordance only, which is at best an indirect and limited measure of “replication”. What about testing their association in psychiatric cohorts?

The Reviewer is correct that we validated new condFDR loci by testing for en masse sign concordance in independent disorder summary statistics. This is a commonly used method in the field (see e.g. see Bansal et al. (2018), Hindley et al. (2022), Torgersen et al. (2022)). Formally testing replication of condFDR effects would require two (disorder, neuroimaging) independent largely sampled cohorts: please note that independent cohorts are needed for each step of this analysis (condFDR, replication, pleioPRS), which limits the available data we can use for replication. We agree that this is a suboptimal solution and have added limitation to the main text. However, we stress that the main purpose of this replication step is to compare sign concordance of the condFDR discovers vs the loci discovered in the original disorder GWAS. Both groups of loci suffer from the limited interpretability of sign concordance compared to direct replication, making it a fair comparison.

Methods, page 21, line 620

Since the probability of replicating these original and conditioned loci at genome-wide significance is low due to the limited statistical power of replication GWAS, we first tested for en masse sign concordance of effect direction in independent summary statistics, as is in line with previous literature^{5,6}. All summary statistics that were used to look up lead SNPs and test for sign concordance are listed in Supplementary Data 12. An exact binomial test was used to test the null hypothesis that

directions of effects were randomly distributed ($prob.=0.5$), given the total number of variants and the number of variants with concordant effects.

5. Minor: (Supplementary Data 16) P4 of the supplementary does not point to the correct table

We thank the Reviewer for pointing out this typo which we have now corrected in the updated version of the manuscript.

6. Is the ABCD mixed ancestry sample composed of a single admixture population or a mix of several populations? In the latter case, are GWAS analyses conclusive? A genetic PC plot could be informative.

We apologize for the unclear description of the ABCD sample. In fact, both of the population characteristics suggested by the Reviewer are true for the ABCD mixed ancestry sample as used in our submitted manuscript: it is a mixture of samples that overlap with one of the 1000 Genomes populations and of samples that have an admixed ancestral background. Fan et al. (2023)⁷ recently described this in detail in a special issue of Behavioral Genetics, including genetic PC plots. We here include Figure 1 from the paper with a genetic PC plot. We have added this population description to our main text.

Legend from Fan et al.
Population structures of ABCD data. First three genetic principal components derived from performing PC-AiR on ABCD genotype data were plotted pairwise, with both ABCD and 1000 Genome Project phase 3 data projected onto the same PC space. The reference groups from the 1000 genome project phase 3 were defined in continental categories and color labeled accordingly. The gray cross marks were ABCD data. (AFR African; AMR Native Americans; EAS East Asian; EUR European; SAS South Asian)

Results, page 4, line 115

We also included the ABCD cohort which had identical sMRI- and dMRI-derived measures and similar fMRI-derived measures (see Methods). The ABCD sample has a heterogeneous and admixed ancestral background. We created a replication sample within ABCD similar to our UKB discovery sample by selecting the subset of European individuals ($N_{\text{sMRI}} = 4,794$, $N_{\text{fMRI}} = 4,132$, $N_{\text{dMRI}} = 4,418$) assigned based on genetic ancestry factor⁸ as defined in Methods. All quality controlled ABCD samples ($N_{\text{sMRI}} = 8,607$, $N_{\text{fMRI}} = 7,277$, $N_{\text{dMRI}} = 7,853$) served as an additional (ad)mixed ancestral replication sample. Next to genetic ancestry, ABCD samples were used to test the generalizability of our results across age.

We would also like to address the second question of the Reviewer, which is certainly valid and a rising challenge in literature given the increasing availability of admixed population samples. Currently most studies – including us – attempt to control for false positive hits due to alleles being at different frequencies across populations by using genetic principal components in a linear model framework⁹. A limitation of genetic principal components is that they don't account for individuals' local ancestry makeup, which may still differ between samples even if their global characteristics are very similar. Using our approach for discovery purposes may therefore not be strict enough. However, our association analysis in the ABCD cohort represents a *secondary* generalization effort: we don't focus on the novel loci discovered in this sample, but report *p*-values of loci identified in the UKB-based discovery analysis. Therefore, we would argue this methodology is of limited concern. We have added it as a limitation in our discussion.

Discussion, page 13, line 347

The results in the ABCD cohort showed generalizability of structural-diffusion MRI pleiotropy from old age to late childhood, and from European ancestry to mixed ancestries. Note that more sophisticated methods for analyses with admixed individuals are becoming available⁹, that allow to model individuals' local ancestry.

7. The modality specific loci could be just due to a lack of power or the way IDPs are grouped – e.g. signal too region specific to show up in MOSTest. I would tone down some of the claims/discussions about these “modality specific” hits.

MOSTest may indeed miss certain hits when genetic signal is sparse across phenotypes, as the reviewer suggests and as pointed out in the original MOSTest paper (Van der Meer et al., 2020). Additionally, we agree that describing certain hits as modality-specific is indeed subject to power. We have rephrased “modality-specific” to “single-modality” throughout the current version of the manuscript to ensure that we are not making claims about specificity but are building our discussion on current association results. We have also enhanced our statement in the Discussion that single-modality term does not claim specificity and is dependent of the grouping of phenotypes and statistical power.

Discussion, page 14, line 382

Some limitations are worth noting when interpreting our results. First, despite our efforts to harmonize the three sets of phenotypes to the greatest degree, the differential spatial granularity and number of features across modalities can result in differential representation of certain brain regions or brain characteristics in the multivariate signal. Second, our definition of single-modality loci and genes is inherent to two factors: 1) the grouping of phenotypes in unimodal and multimodal analyses – MOSTest may miss a small number of hits when genetic signal is sparse across the jointly analyzed phenotypes¹⁰,

and 2) the currently available data limits our statistical power – loci that are now associated with one modality could become genome-wide significant in another modality once sample sizes increase.

Discussion, page 13, line 359

Our enrichment analyses showed that the biological processes and molecular components ~~can be decomposed to those 1) enriched for either sMRI specific (neuron development, differentiation, and migration, the synapse, axon) or dMRI specific genes (cell death, cellular response to stimuli), 2) converging from modality specific and cross-modality genes (regulation of neuron/cell projection development, organization and guidance, microtubule), and 3) only enriched of cross-modality genes~~ represent general functions such as cell cycle processes, regulation of gene expression, cell junctions. That gene-sets implicated by cross-modality genes alone are involved in fundamental biological processes is consistent with previous findings that genes associated with multiple trait domains are more likely to be involved in general biological functions¹¹. This is in line with our finding that, although cross-modality genes were highly expressed in fetal brain tissue, those genes were not necessarily cell-specific – suggesting they may serve in cellular functions taking place in all cell-types. Future research may explore this putative relationship between the degree of pleiotropy and the specificity of biological functions.

8. *Sign concordance tests in independent disorder summary statistics. Sorry, I did not get why you are not relying on actual replication tests?*

This comment was already addressed in the response to a previous comment by the reviewer on page 15. For clarity, we refer to the same response here:

The reviewer is correct that we validated new condFDR loci by testing for en masse sign concordance in independent disorder summary statistics. This is a commonly used method in the field (see e.g. see Bansal et al. (2018), Hindley et al. (2022), Torgersen et al. (2022)). Formally testing replication of condFDR effects would require two (disorder, neuroimaging) independent largely sampled cohorts: please note that independent cohorts are needed for each step of this analysis (condFDR, replication, pleioPRS), which limits the available data we can use for replication. We agree that this is a suboptimal solution and have added limitation to the main text. However, we stress that the main purpose of this replication step is to compare sign concordance of the condFDR discovers vs the loci discovered in the original disorder GWAS. Both groups of loci suffer from the limited interpretability of sign concordance compared to direct replication, making it a fair comparison.

Methods, page 21, line 620

Since the probability of replicating these original and conditioned loci at genome-wide significance is low due to the limited statistical power of replication GWAS, we first tested for en masse sign concordance of effect direction in independent summary statistics, as is in line with previous literature^{5,6}. All summary statistics that were used to look up lead SNPs and test for sign concordance are listed in Supplementary Data 12. An exact binomial test was used to test the null hypothesis that directions of effects were randomly distributed ($prob.=0.5$), given the total number of variants and the number of variants with concordant effects.

9. Stable 14 – $r^2 \gg 1$. R2 and coefficient columns may have been swapped?

We thank the reviewer for pointing out this error. We have checked our result files and the columns had indeed been swapped. We have corrected this in the current revised version of the manuscript.

10. The pleiotropy you reported is limited to common SNPs, and it could be that a different pattern may emerge when looking at rarer variants?

Our manuscript indeed focusses on common variants. We have now added a section to our Discussion that describes the level of multimodal pleiotropy that could be expected for rare variants.

Discussion, page 13, line 338

It is expected from theoretical analysis that rare variants would follow this pattern of pleiotropy under assumption of the infinitesimal model¹², which would be an interesting direction of future research given the recent availability of multivariate tools for gene-based rare-variant association studies^{13,14}.

Reviewer #2

Tissink et al. conducted multivariate GWAS on multiple phenotypes of different MRI modalities. They identified pleiotropic genetic associations and discussed some biological interpretations. Identified candidate genes appear to be enriched in prenatal brain development. The multimodal analysis can also assist genetic mapping in psychiatric disorders. The analysis in general is interesting, and I have the following major concerns or comments.

Regarding the results:

1. Generally, multivariate analysis is always tricky to interpret. I very much hope that the authors can try to dig further into the multivariate associations and identify the genotype-phenotype maps in a more clear manner. Where did the multivariate power come from? Which trait particularly are the SNPs affecting, especially for the cross-modality associations? It would be much better to have some illustrations of the genetic effects directions so that we understand the genetic correlations caused by the pleiotropic loci better.

We thank the reviewer for giving us the opportunity to present more detailed results. First, we have enriched our Introduction with the rationale why multivariate analyses are more powerful than univariate analyses in the presence of widespread pleiotropy.

Introduction, page 4, line 64

Complex traits are affected by thousands of genetic variants scattered throughout the genome which makes pleiotropy between them inevitable¹⁵. In presence of such abundant pleiotropy, multivariate GWAS models have greater statistical power than univariate GWAS models¹⁶. This is because the cumulative evidence from different phenotypes leads to more sensitive detection of genetic associations and reduces the burden of multiple testing¹⁰.

Second, we have now added a new main Figure (1c) that illustrates how the univariate statistics (z-scores) contribute to the multivariate statistics for 10 example lead SNPs from the Venn Diagram (Figure 1a). From this Figure, it is clear that the associations boosted by the multimodal MOSTest

analysis (yellow) are not driven by any particular trait, but arise due to aggregation of multiple sub-threshold univariate p -values.

Results, page 7, line 188

We examined the univariate associations underlying multivariate associations from different parts of the Venn diagram (Fig. 1a). Fig. 1c shows univariate z-scores across sMRI-, fMRI-, and dMRI-derived phenotypes for ten lead SNPs representing different parts of the Venn diagram (Fig. 1a). Cluster map of univariate z-scores for all multimodal MOSTest lead SNPs is displayed in Supplementary Fig. 6. Alternatively, for every lead SNP identified in the MOSTest analyses presented in the Venn diagram (Fig. 1a) we extracted the minimum univariate p -value across all analysed phenotypes. A lead SNP with a relatively high minimum univariate p -value would indicate that the signal should have been highly distributed across other measures for the variant to become genome-wide significant in the multivariate analysis. Supplementary Fig. 7 shows that lead SNPs boosted by the multimodal analysis had relatively high minimum univariate p -values ($0.05 > p > 5 \times 10^{-5}$) more frequently (87%) than lead SNPs of loci identified in one unimodal analysis only (sMRI-modality 72%; fMRI-modality 25%; dMRI-modality 78%), suggesting that the discovery of these lead SNPs is driven by pleiotropic signals across modalities.

Results, page 6, line 158

c) Univariate z-scores for multivariate lead SNPs

Figure 1c. For 10 example lead SNPs from different parts of a) the univariate z-scores of all phenotypes used in this study (sMRI-derived phenotypes in green, dMRI-derived phenotypes in red, fMRI-derived phenotypes in blue) are plotted in c) as the distance from the centre of each circular plot. The dashed black line corresponds to $p = 5 \times 10^{-8}$ ($z = 5.45$) and z-scores that pass this threshold are depicted larger. Positive effect direction is shown as a filled circle and negative effect direction by a white circle.

We have also attempted to go beyond these example SNPs and create a large matrix with the univariate statistics (z-scores) for all multimodal MOSTest lead SNPs (as a generalization of Figure 1c) in Supplementary Figure 6. Given the number of phenotypes and lead SNPs, it has been challenging to present the effects clearly. We attempted to aid interpretation by applying clustering and observe that phenotypes from the same modality tend to cluster in groups (column clustering), while variants from different sectors of Venn diagram are mixed (row clustering).

Supplementary Figure 6. Univariate p -values underlying all lead SNPs identified through multivariate GWAS in MOSTest. Lead SNPs are categorized by their location in the Venn diagram (Fig. 1a in main text) on the y-axis, phenotypes are clustered on the x-axis. Hierarchical clustering was applied (Methods).

2. *The functional consequences of the lead SNPs using VEP seem striking to me. It appears to be common sense that most GWAS associations sit on non-coding regions, normally centered around transcription start sites of genes. Due to LD, one would expect that most of the lead SNPs would be in the non-coding regions or synonymous, and very few would be missense/non-synonymous. Thus for a GWAS analysis, I would not expect a significant enrichment of missense variants as shown in Figure 2. How about changing the background from random sets of pruned SNPs to GWAS hits of other types of human complex traits? This is asking whether the observed functional consequences distributions are special for the MRI phenotypes.*

We agree with the reviewer that most GWAS associations sit on non-coding regions. In fact, a large overview study by Watanabe et al. (2020) described that 93.1% of the SNPs in loci from 558 well-powered traits are non-coding. Nevertheless, since the majority of the genome is non-coding, this still resulted in a significant depletion of intergenic SNPs (under assumption of uniform distribution of

associations in the genome). We also agree with the reviewer that, looking at lead SNPs only, very few would be expected to be missense/non-synonymous. However, because a small proportion of the genome is missense/non-synonymous to start with, this could still result in a significant enrichment. This is also what Watanabe et al. (2020) observed: the strongest enrichment was seen in exonic lead SNPs, while intergenic regions were depleted. Similar observations have been made in other overview papers, e.g. by Schork et al. (2013). Our results were therefore similar to the enrichment patterns reported for other complex traits. We have revised the Figure to now display the proportion of SNPs on the y-axis (instead of $\log_2(\text{OR})$) as this may be more intuitive, in addition to revising the legend text of this Figure to clarify this issue.

We also thank the reviewer for the suggestion to use an alternative background for the enrichment analysis and agree it would be interesting to find out what functional consequences are special for MRI phenotypes. We have therefore leveraged the information available from Watanabe et al. (2020). The authors of that study collected 4,155 GWASs from 295 unique studies covering 2,965 unique traits. Subsequently, they selected the GWAS with the largest sample size ($N > 50,000$) per trait, resulting in 558 GWASs for 558 unique traits across 24 trait domains. They annotated the 43,492 unique lead SNPs from these traits, and provide the proportions of ANNOVAR annotations in Table 2. We have now used this information as the reference of our enrichment analysis. The results are rather interesting, given that cross-modality lead SNPs showed unique enrichment of exonic variants. Because Watanabe et al. demonstrate that the contribution of exonic SNPs increases from ~1% (SNPs associated to 1 trait domain) to ~5% (associated to ≥ 10 trait domain), the enrichment we observed emphasizes the strong pleiotropic nature of the cross-modality lead SNPs.

Results, page 8, line 202

We investigated to what extent single-modality (identified in only one unimodal analysis) and cross-modality (identified in ≥ 2 unimodal analyses or boosted by the multimodal analysis) loci and genes differ in their biological effects (Supplementary Data 7-8) compared to other complex traits. To this end, we annotated 251 sMRI-, 11 fMRI-, 177 dMRI-modality, and 462 cross-modality lead SNPs using ANNOVAR² (Supplementary Data 9). We used the annotations of 43,492 (unique) lead SNPs derived from 558 traits (with reasonable power, i.e. $N > 50,000$) across 24 trait domains from Watanabe et al.¹¹ as a reference. Fig. 3b demonstrates that cross-modality and single-modality lead SNPs were very similarly located in genomic regions as other complex traits^{11,17}. Notably, cross-modality lead SNPs showed unique enrichment of exonic variants (6.58%, OR = 2.17, 95% CI = 1.43-3.17, $p = 2.91 \times 10^{-4}$). Watanabe et al. already demonstrated that the contribution of exonic SNPs increases from ~1% (SNPs associated to 1 trait domain) to ~5% (associated to ≥ 10 trait domain)¹¹. The enrichment we observed therefore emphasizes the strong pleiotropic nature of the cross-modality lead SNPs.

Figure 2. b) Functional consequences of single-modality and cross-modality lead SNPs as annotated with ANNOVAR². When the null hypothesis (OR=1) could be rejected after Bonferroni correction ($p < 1.14 \times 10^{-3}$), the solid line indicates significant enrichment. Annotations of 43,492 (unique) lead SNPs derived from 558 traits across 24 trait domains from Watanabe et al. were used as reference for Fisher Exact Test (Supplementary Data 9).

Methods, page 19, line 571

We were interested in potential differences between single-modality and cross-modality loci compared to other complex traits. Therefore we selected the lead SNPs within the respective loci (see *Locus definition*) and annotated them with ANNOVAR². As reference, we used the annotations of 43,492 (unique) lead SNPs derived from 558 traits (with reasonable power, i.e. $N > 50,000$) across 24 trait domains (Table 2 from Watanabe et al.¹¹). Enrichment of the single-modality and cross-modality lead SNPs in positional annotation categories was then tested using Fisher's Exact test. The alpha level for significant enrichment was Bonferroni corrected ($\alpha = 0.05 / 44 = 1.14 \times 10^{-3}$).

3. The lifespan expression patterns in Figure 3b look interesting. Is there any other way to validate the cross-modality genes identified in terms of this perspective? To what extent can we justify their roles in fundamental biological processes and prenatal brain development?

We have used single-cell RNA sequencing data from Bhaduri et al.¹⁸ in fetal brain tissue to perform a cell-type enrichment analysis for the single-modality and cross-modality genes to validate the general role cross-modality genes have in terms of biological functions during prenatal brain development. We

observed that cross-modality genes were enriched across the majority of cell types, confirming the importance of these genes in fetal brain tissue. Though we did not observe specificity for one cell-type, suggesting cross-modality genes may serve in cellular functions taking place in many or all cell-types.

Methods, page 20, line 600

Lastly, we explored cell-type enrichment of single-modality and cross-modality genes by performing Fisher Exact Tests for each cell-type identified by Bhaduri et al.¹⁸ using single-cell RNA sequencing analyses of the fetal brain. The minimum number of genes overlapping was set to >2. Bonferroni correction was applied for the number of tests performed ($\alpha = 0.05 / 30 = 1.67 \times 10^{-3}$).

Results, page 9, line 237

Figure 3. d) Cell-type enrichment analysis for fetal tissue from Bhaduri et al.¹⁸. Bonferroni corrected significant results ($p < (0.05 / 30) = 1.67 \times 10^{-3}$) are indicated by an asterisk (*). ipc = intermediate progenitor cells, oligo = oligodendrocytes.

Results, page 8, line 235

Given the predominant prenatal gene-expression pattern, we investigated whether single-modality and cross-modality genes were enriched for any cell-type identified by Bhaduri et al.¹⁸ using single-cell RNA sequencing analysis of the fetal brain (Fig. 2d). We observed that cross-modality genes were enriched across all cell-types, confirming their importance in fetal brain tissue and suggesting their expression pattern is not cell-specific and more general in nature (Supplementary Data 11).

Discussion, page 13, line 360

Our enrichment analyses showed that the biological processes and molecular components only enriched for cross-modality genes include more general functions such as cell cycle processes, regulation of gene expression, cell junctions. That gene-sets implicated by cross-modality genes alone are involved in fundamental biological processes is consistent with previous findings that genes associated with multiple trait domains are more likely to be involved in general biological functions¹¹. This is in line with our finding that, although cross-modality genes were highly expressed in fetal brain tissue, those genes were not necessarily cell-specific – suggesting they may serve in cellular functions taking place in all cell-types. Future research may explore this putative relationship between the degree of pleiotropy and the specificity of biological functions.

4. Regarding the shared genetic architecture with psychiatric disorders, the authors focused on genetic discovery conditioned on the MRI multivariate associations. The idea is fine, but I don't think the results have been shown clearly. I expect e.g., an additional main figure showing 1) the boost of discovery power, 2) the new discoveries conditioned on loci from different MRI modalities; 3) what have we

missed in the standard GWAS for psychiatric disorders, including heritability and biological mechanisms?

We thank the reviewer for the opportunity to present our results more clearly. We have now added a new main figure that shows 1) the boost in discovery, 2) the PGS performance for disorders once conditioned on multimodal MRI summary statistics, and 3) new results that show that conditioning on multivariate brain GWAS substantially enriches the "brain-related" component of the disorders. Exploring specific biological mechanisms per disorder is out of scope for this study and should thoroughly be investigated in separate studies focused on these psychiatric traits. Standard methods for heritability assessment (e.g. LDSC) rely on p -value (uniformly distributed under null hypothesis of no association) and thus are not applicable to FDR values estimated by conjFDR analysis. We therefore focussed on what conditioning on multimodal summary statistics does to the tissue-specificity of psychiatric disorder genes, for which we used positional gene mapping and hypergeometric test-based tissue enrichment analyses in FUMA.

Methods, page 21, line 628

Tissue enrichment of multimodal | disorder genes and original disorder genes

Using the genome-wide significant loci from the condFDR and original summary statistics, we used FUMA¹⁹ (<https://fuma.ctglab.nl/>) to positionally map SNPs to genes. The resulting gene lists were used for hypergeometric test-based tissue enrichment analyses (as implemented in FUMA), to test the tissue-specificity of the input genes for differentially expressed (upregulated) genes in each of the 54 tissues available in GTEx (v8)²⁰. Bonferroni correction was applied for the number of tests performed ($\alpha = 0.05 / (54 \text{ tissues} \times 2 \text{ gene lists} \times 5 \text{ disorders}) = 9.26 \times 10^{-5}$).

Results, page 10, line 273

We positionally mapped condFDR loci to genes (528 genes for MDD, 2,416 for SCZ, 116 for ASD, 1,162 for BD, 122 for ADHD; Supplementary Data 15) and subsequently used these genes as input for hypergeometric test-based tissue enrichment analyses, both as implemented in FUMA. The SCZ and BD genes were significantly enriched in gene-sets with upregulated gene expression in brain tissue, but not in other body tissues (Supplementary Data 16, Fig. 3c). This contrasts with the sets of genes mapped from the original GWAS loci (60 genes for MDD, 498 for SCZ, 5 for ASD, 186 for BD, 14 for ADHD; Supplementary Data 15), which showed no enrichment of upregulated gene expression in brain tissues. This illustrates that genetic overlap may be used to identify genetic variants that may play a role in psychiatric disorders with a specific relevance for brain traits.

Figure 3. a) Loci discovered in the publicly available summary statistics of each disorder GWAS ($p < 5 \times 10^{-8}$), and after conditioning the disorder on multimodal MOSTest summary statistics with conditional FDR ($FDR < 0.05$). b) Phenotypic variance explained for five psychiatric disorders explained by polygenic scores based on original disorder GWAS summary statistics and conditional disorder-multimodal GWAS summary statistics. Note that PGS using top 10 and top 100 SNPs had too low R^2 values to be visible and are therefore not plotted, but can be found in Supplementary Data 17. Independent target samples were used for prediction (see Methods). A random sampling procedure and t -test were applied per disorder to test if the observed difference in R^2 was significant (Supplementary Fig. 9). c) Enrichment of disorder genes (mapped from the original disorder GWAS loci vs conditional FDR loci) in tissue-specific gene-sets with upregulated expression (Supplementary Data 16). The red line indicates Bonferroni correction for the number of tests ($\alpha = 0.05/540 = 9.26 \times 10^{-5}$).

5. *Following the above, how do the findings of this study relate to or build upon previous research on genetic pleiotropy in the context of neuroimaging and psychiatric disorders? Are there any inconsistencies or discrepancies that warrant further investigation?*

We have now added a new section to the Introduction and Discussion that describe the previous knowledge on genetic pleiotropy in the context of neuroimaging and psychiatric disorders and how our study adds novel insight and follow-up research questions.

Introduction, page 3, line 79

A large body of neuroimaging research established associations between psychiatric disorders and brain structure^{21–26}, function^{27,28} and diffusion metrics^{29,30}. However, previous studies were not able to detect non-null genetic correlations^{31–33}, possibly due to mixed effect directions³⁴ and differential polygenicity between neuroimaging phenotypes and psychiatric disorders³⁵. Alternative methods that leverage the shared genetic signal between brain morphology and psychiatric disorders have been more fruitful³⁴ allowing improved prediction of disease liability³. Given that alterations of brain morphology, activity, connectivity, and tissue composition often co-occur in heritable psychiatric disorders^{36,37}, using multimodal GWAS associations may potentially further aid in leveraging the shared genetic signal between neuroimaging traits and psychiatric conditions.

Discussion, page 14, line 376

When conditioning GWAS summary statistics for five major psychiatric disorders on our multimodal multivariate analysis, we discovered new loci including genes with increased brain-specific gene expression. The validity of these boosted loci needs to be replicated in independent samples in future research. Nevertheless, prioritizing these variants already showed enhanced polygenic prediction for MDD, BD, ASD, and ADHD. This highlights that future diagnostic strategies targeting psychiatric disorders may benefit from aligning the PGSs to a relevant endophenotype. This rationale is similar to a previously published study that showed improved prediction of SCZ after filtering on genetic variants expressed in the placenta⁴. Exploring which disorders benefit from either a more specific or broader range of neuroimaging phenotypes, could also be an intriguing area for future research.

6. *Are there any potential limitations or biases in the gene expression data used to determine the temporal and spatial specificity of the cross-modality genes, and how might these limitations impact the interpretation of the results?*

We thank the Reviewer for the opportunity to critically discuss our temporal gene-expression results. We have added a limitation to our Discussion that discusses the biases in gene expression data obtained from *postmortem* brain tissue.

Discussion, page 14, line 385

Some limitations are worth noting when interpreting our results. ... Fifth, the temporal gene-expression patterns of single-modality and cross-modality genes are based on a *postmortem* dataset with a limited number of donors per timepoint³⁸. Moreover, a study that was recently released compared gene-expression between *postmortem* and living brain samples and found a significant difference in 80% of the genes³⁹. Although living postnatal gene-expression samples may become available in the future for validation of our findings, prenatal gene-expression will remain to rely on *postmortem* brain tissue.

Regarding the methods:

7. Regarding the multivariate test method, how well does the Multivariate Omnibus Statistical Test (MOSTest) account for potential confounding factors, such as population stratification? Genomic PCs were included as covariates in the univariate analysis, but I'm not sure whether that is sufficient for the multivariate analysis.

We thank the reviewer for this important comment and agree with the reviewer that multivariate analysis can exacerbate inflation introduced through population stratification if it is only partly controlled in the univariate GWAS analysis. We try to minimize potential inflation due to population stratification by limiting our UKB-based discovery analysis to genetically homogeneous white British population (individuals who self-identified as white British and have very similar genetic ancestry based on a genetic PC analysis) and additionally by accounting for the first 20 genetic principal components. In our case all univariate GWAS performed (subsequently combined in MOSTest) had an LD Score Regression intercept below 1.03, indicating no substantial inflation due to population stratification or cryptic relatedness. Moreover, if the multivariate GWAS signal would have been largely driven by population stratification, one would not expect to find biologically meaningful results, or consistent findings in the replication analysis within the ABCD cohort. Lastly, this study follows GWAS analytical pipeline which was developed and validated in the previously published studies focused on a single imaging modality (sMRI: van der Meer et al. (2020), dMRI: Fan et al. (2022), fMRI: Roelfs et al. (2022)).

To further support that ancestry has been appropriately controlled for, we have performed an additional analysis that allowed us to inspect whether any remaining ancestry clusters would appear from our lead SNPs. Looking at the first four genetic principal components, we can conclude that the genetic variance in the SNPs that were found to be associated with multimodal neuroimaging-derived phenotypes were not driven by population stratification. Moreover, we applied LD Score Regression to the MOSTest summary statistics and concluded that the intercept of 1.04 (0.04) indicates that the inflation in multimodal MOSTest p -values ($\lambda_{GC} = 2.83$) was driven by polygenicity and not population stratification or cryptic relatedness.

Results, page 7, line 170

We next investigated whether combining all sMRI, fMRI, and dMRI-derived measures in one multivariate analysis generated greater statistical power to identify novel pleiotropic loci and genes which show sub-threshold associations in each unimodal multivariate analysis (Manhattan plot in Supplementary Fig. 3). We therefore applied MOSTest across neuroimaging modalities, combining 583 phenotypes, identifying 851 genetic loci (Supplementary Data 2; replication rates EUR 26.45%, 37.55% mixed ancestries; Supplementary Data 6). The LD Score Regression intercept (1.04, SE=0.04) indicated

that the inflation in these multimodal MOSTest p -values ($\lambda_{GC} = 2.83$) was driven by polygenicity and not population stratification or cryptic relatedness.

8. *Given the sample sizes of the UK Biobank and ABCD study, can the authors address the potential impact of statistical power on the robustness of their findings, and perhaps discuss the sample size required in the future to better reveal the genetic basis of MRI phenotypes?*

We thank the reviewer for this comment. The question about the required sample size for well-powered multivariate GWAS studies using neuroimaging was previously investigated. We have now added a section to the Discussion where we address this topic. Additionally, we have clarified that increased statistical power in the future would most likely affect our definition of which genes or loci are shared across modalities and which are not. We have therefore toned down our wording and no longer refer to “modality-specific” effects, because this is subject to power.

Discussion, page 13, line 349

A lack of power in the smaller ABCD cohort most likely limited the robust estimation of genetic associations for the relatively low heritable functional MRI-derived phenotypes, which complicated examining the generalizability of structural-functional-diffusion MRI pleiotropy beyond structural-diffusion MRI pleiotropy. The necessary future samples size to uncover the heritability of neuroimaging-derived phenotypes depends on genetic architecture and varies across modalities or the brain trait of interest⁴⁰. For example, researchers expect that multivariate analyses of the prospective UK Biobank neuroimaging sample of 100,000 individuals will explain 32% and 24% of the additive genetic variance in cortical surface area and thickness respectively⁴¹. This highlights that both larger neuroimaging-genetic datasets and novel statistical approaches are required to delineate genetic background of neuroimaging phenotypes.

Discussion, page 14, line 388

Second, our definition of single-modality findings (only found in one unimodal analysis) is inherent to two factors. 1) The grouping of phenotypes in unimodal and multimodal analyses: MOSTest may miss a small number of hits when genetic signal is sparse across the jointly analyzed phenotypes¹⁰. 2) The currently available data limits our statistical power: loci that are now associated with one modality could also become genome-wide significant in another modality once sample sizes increase.

Regarding the discussions:

9. *Can the authors discuss the implications of their findings for the development of future diagnostic and therapeutic strategies targeting psychiatric disorders, and highlight potential areas for further research?*

We have now added a section to the Discussion where we elucidate on the implications of our findings for the development of future diagnostic strategies targeting psychiatric disorders using polygenic scores, including potential areas for further research.

Discussion, page 14, line 376

When conditioning GWAS summary statistics for five major psychiatric disorders on our multimodal multivariate analysis, we discovered new loci including genes with increased brain-specific gene

expression. The validity of these boosted loci needs to be replicated in independent samples in future research. Nevertheless, prioritizing these variants already showed enhanced polygenic prediction for MDD, BD, ASD, and ADHD. This highlights that future diagnostic strategies targeting psychiatric disorders may benefit from aligning the PGSs to a relevant endophenotype. This rationale is similar to a previously published study that showed improved prediction of SCZ after filtering on genetic variants expressed in the placenta⁴. Exploring which disorders benefit from either a more specific or broader range of neuroimaging phenotypes, could also be an intriguing area for future research.

References

1. de Leeuw, C. A., Mooij, J. M., Heskes, T. & Posthuma, D. MAGMA: Generalized Gene-Set Analysis of GWAS Data. *PLoS Comput. Biol.* **11**, 1–19 (2015).
2. Wang, K., Li, M. & Hakonarson, H. ANNOVAR: Functional annotation of genetic variants from high-throughput sequencing data. *Nucleic Acids Res.* **38**, 1–7 (2010).
3. van der Meer, D. *et al.* Boosting Schizophrenia Genetics by Utilizing Genetic Overlap With Brain Morphology. *Biol. Psychiatry* **92**, 291–298 (2022).
4. Ursini, G. *et al.* Convergence of placenta biology and genetic risk for schizophrenia. *Nat. Med.* **24**, 792–801 (2018).
5. Bahrami, S. *et al.* Dissecting the shared genetic basis of migraine and mental disorders using novel statistical tools. *Brain* **145**, 142–153 (2022).
6. Torgersen, K. *et al.* Shared genetic loci between depression and cardiometabolic traits. *PLoS Genet.* **18**, 1–25 (2022).
7. Fan, C. C. *et al.* Genotype Data and Derived Genetic Instruments of Adolescent Brain Cognitive Development Study® for Better Understanding of Human Brain Development. *Behav. Genet.* 159–168 (2023). doi:10.1007/s10519-023-10143-0
8. Raj, A., Stephens, M. & Pritchard, J. K. FastSTRUCTURE: Variational inference of population structure in large SNP data sets. *Genetics* **197**, 573–589 (2014).
9. Atkinson, E. G. *et al.* Tractor uses local ancestry to enable the inclusion of admixed individuals in GWAS and to boost power. *Nat. Genet.* **53**, 195–204 (2021).
10. van der Meer, D. *et al.* Understanding the genetic determinants of the brain with MOSTest. *Nat. Commun.* **11**, (2020).
11. Watanabe, K. *et al.* A global overview of pleiotropy and genetic architecture in complex traits. *Nat. Genet.* **51**, 1339–1348 (2019).
12. Turelli, M. Heritable genetic variation via mutation-selection balance: Lerch’s zeta meets the abdominal bristle. *Theor. Popul. Biol.* **25**, 138–193 (1984).
13. Luo, L. *et al.* Multi-trait analysis of rare-variant association summary statistics using MTAR. *Nat. Commun.* **11**, 1–11 (2020).
14. Dutta, D., Scott, L., Boehnke, M. & Lee, S. Multi-SKAT: General framework to test for rare-variant association with multiple phenotypes. *Genet. Epidemiol.* **43**, 4–23 (2019).
15. Sivakumaran, S. *et al.* Abundant pleiotropy in human complex diseases and traits. *Am. J. Hum. Genet.* **89**, 607–618 (2011).
16. Zhu, W. & Zhang, H. Why do we test multiple traits in genetic association studies? *J. Korean Stat. Soc.* **38**, 1–10 (2009).
17. Schork, A. J. *et al.* All SNPs are not created equal: Genome-wide association studies reveal a consistent pattern of enrichment among functionally annotated SNPs. *PLoS Genet.* **9**, (2013).
18. Bhaduri, A. *et al.* An atlas of cortical arealization identifies dynamic molecular signatures. *Nature* **598**, 200–204 (2021).
19. Watanabe, K., Taskesen, E., van Bochoven, A. & Posthuma, D. Functional mapping and annotation of genetic associations with FUMA. *Nat. Commun.* **8**, 1826 (2017).
20. The GTEx Consortium. The GTEx Consortium atlas of genetic regulatory effects across human tissues. *Science (80-.).* **369**, 1318–1330 (2020).
21. Boedhoe, P. S. W. *et al.* Subcortical brain volume, regional cortical thickness, and cortical surface area across disorders: Findings from the ENIGMA ADHD, ASD, and OCD working

- groups. *Am. J. Psychiatry* **177**, 834–843 (2020).
22. Hibar, D. P. *et al.* Cortical abnormalities in bipolar disorder: An MRI analysis of 6503 individuals from the ENIGMA Bipolar Disorder Working Group. *Mol. Psychiatry* **23**, 932–942 (2018).
 23. van Erp, T. G. M. *et al.* Cortical Brain Abnormalities in 4474 Individuals With Schizophrenia and 5098 Control Subjects via the Enhancing Neuro Imaging Genetics Through Meta Analysis (ENIGMA) Consortium. *Biol. Psychiatry* **84**, 644–654 (2018).
 24. van Erp, T. G. M. *et al.* Subcortical brain volume abnormalities in 2028 individuals with schizophrenia and 2540 healthy controls via the ENIGMA consortium. *Mol. Psychiatry* **21**, 547–553 (2016).
 25. Schmaal, L. *et al.* Cortical abnormalities in adults and adolescents with major depression based on brain scans from 20 cohorts worldwide in the ENIGMA Major Depressive Disorder Working Group. *Mol. Psychiatry* **22**, 900–909 (2017).
 26. Schmaal, L. *et al.* Subcortical brain alterations in major depressive disorder: findings from the ENIGMA Major Depressive Disorder working group. *Mol. Psychiatry* **21**, 806–812 (2016).
 27. van den Heuvel, M. P. & Sporns, O. A cross-disorder connectome landscape of brain dysconnectivity. *Nat. Rev. Neurosci.* (2019). doi:10.1038/s41583-019-0177-6
 28. Rudie, J. D. *et al.* Altered functional and structural brain network organization in autism. *NeuroImage Clin.* **2**, 79–94 (2013).
 29. Daianu, M. *et al.* Disrupted rich club network in behavioral variant frontotemporal dementia and early-onset Alzheimer’s disease. *Hum. Brain Mapp.* **37**, 868–883 (2016).
 30. Kelly, S. *et al.* Widespread white matter microstructural differences in schizophrenia across 4322 individuals: Results from the ENIGMA Schizophrenia DTI Working Group. *Mol. Psychiatry* **23**, 1261–1269 (2018).
 31. Franke, B. *et al.* Genetic influences on schizophrenia and subcortical brain volumes: Large-scale proof of concept. *Nat. Neurosci.* **19**, 420–431 (2016).
 32. Tisink, E. *et al.* Genome-wide association study of cerebellar volume provides insights into heritable mechanisms underlying brain development and mental health. *Commun. Biol.* **5**, (2022).
 33. Ohi, K. *et al.* Genetic correlations between subcortical brain volumes and psychiatric disorders. *Br. J. Psychiatry* **216**, 280–283 (2020).
 34. Cheng, W. *et al.* Shared genetic architecture between schizophrenia and subcortical brain volumes implicates early neurodevelopmental processes and brain development in childhood. *Mol. Psychiatry* (2022). doi:10.1038/s41380-022-01751-z
 35. Cheng, W. *et al.* Genetic Association Between Schizophrenia and Cortical Brain Surface Area and Thickness. *JAMA Psychiatry* 1–11 (2021). doi:10.1001/jamapsychiatry.2021.1435
 36. Cross-Disorder Group of the Psychiatric Genomics Consortium. Genomic Relationships, Novel Loci, and Pleiotropic Mechanisms across Eight Psychiatric Disorders. *Cell* **179**, 1469–1482 (2019).
 37. Liu, S. *et al.* Multimodal neuroimaging computing: a review of the applications in neuropsychiatric disorders. *Brain Informatics* **2**, 167–180 (2015).
 38. Kang, H. J. *et al.* Spatio-temporal transcriptome of the human brain. *Nature* **478**, 483–489 (2011).
 39. Liharska, L. E. *et al.* A study of gene expression in the living human brain Living Brain Project cohort. *bioRxiv* (2023).
 40. van der Meer, D. *et al.* The genetic architecture of human cortical folding. *Sci. Adv.* **7**, (2021).
 41. Shadrin, A. A. *et al.* Vertex-wise multivariate genome-wide association study identifies 780 unique genetic loci associated with cortical morphology. *Neuroimage* **244**, (2021).

Reviewer #1 (Remarks to the Author):

Thank you for your detailed response to comments, and for all the work that went into these revisions. I think that the manuscript has improved, and the additional figures are really informative. I have one concern about the PRS testing, which is a central result in your analysis as it is the way to externally validate the discovery and power boost of the multivariate approach. Please correct me if I misunderstood your statistical testing.

"We applied random sampling without replacement procedure and t-test per disorder to test if the highest R2 (MDD pleioPGS/SCZ original PGS with nSNP = 50,000 SNPs; ADHD/ASD/BD pleioPGS with nSNP = 100,000 SNPs) significantly higher than the R2 of the alternative PGS (MDD original PGS/SCZ pleioPGS with nSNP = 50,000 SNPs; ADHD/ASD/BD original PGS with nSNP = 100,000 SNPs). For each trait and type of PGS (original PGS, pleioPGS), we performed 200 iterations randomly sampling 2/3 of controls and 2/3 of corresponding disorder cases with replacement and applying PRSice2 to estimate R2. Obtained distributions of R2 values were used to assess confidence intervals."

It is not clear to me how you tested the difference in R2, and whether this is an appropriate statistical test. The way I understand the procedure is that you estimated two distributions of R2, by sampling the data 200 times, and used a t-test to compare the mean R2. If this is the case, I don't think this is an appropriate test, as its power depends on the number of sampling (here 200) and not on the sample size. Basically, if you had done 10,000 sampling iterations most t-test between the two distributions would come up significant. Maybe I misunderstood, so can you please clarify how the test was performed?

A more straightforward test would consist in fitting a model with the 2 PRS, and comparing it to a model with only the standard PRS. The likelihood ratio test can be used to compare the two nested models. If you really want confidence intervals around R2, I suggest to use bootstrap. I am also aware of this R package, if you are after a parametric approach. [https://www.cell.com/ajhg/pdf/S0002-9297\(23\)00004-6.pdf](https://www.cell.com/ajhg/pdf/S0002-9297(23)00004-6.pdf)

Overall, the fact that you only observe highly significant differences on only a handful of PRS scenarios make me doubt of the validity of the tests.

Minor:

In your point about power of GWAS : Can you clarify what you mean by additive genetic variance, is it the 32% of the SNP heritability or 32% of the twin based heritability (thus close to the 50% mark that is tagged by common SNPs).

"For example, researchers expect that multivariate analyses of the prospective UK Biobank neuroimaging sample of 100,000 individuals will explain 32% and 24% of the additive genetic variance in cortical surface area and thickness respectively"

Reviewer #2 (Remarks to the Author):

The authors have conducted a careful revision of the manuscript. I only have two remaining questions regarding the new Figure 3:

1) The number of loci discovered was used as the criterion comparing the power of univariate and multivariate analysis, but different thresholds were applied: $p < 5e-8$ for univariate and $FDR < 0.05$ for multivariate. Is this really a fair comparison of the statistical power of the two testing strategies?

2) The text in the figure color keys looks funny. I'm not sure what went wrong, but it also looks so in the rebuttal letter.

Please find below our point-by-point response to the comments raised by the reviewers together with the adjustments we made to the manuscript. The questions and comments of the reviewer are presented in italics, our response in normal typeset. We marked changes to the text in blue in the updated version of our manuscript.

Reviewer #1

Thank you for your detailed response to comments, and for all the work that went into these revisions. I think that the manuscript has improved, and the additional figures are really informative.

Response: We thank the Reviewer for the positive note and have addressed their remaining comments below.

I have one concern about the PRS testing, which is a central result in your analysis as it is the way to externally validate the discovery and power boost of the multivariate approach. Please correct me if I misunderstood your statistical testing.

“We applied random sampling without replacement procedure and t-test per disorder to test if the highest R^2 (MDD pleioPGS/SCZ original PGS with $n_{\text{SNP}} = 50,000$; ADHD/ASD/BD pleioPGS with $n_{\text{SNP}} = 100,000$) significantly higher than the R^2 of the alternative PGS (MDD original PGS/SCZ pleioPGS with $n_{\text{SNP}} = 50,000$; ADHD/ASD/BD original PGS with $n_{\text{SNP}} = 100,000$). For each trait and type of PGS (original PGS, pleioPGS), we performed 200 iterations randomly sampling 2/3 of controls and 2/3 of corresponding disorder cases with replacement and applying PRSice2 to estimate R^2 . Obtained distributions of R^2 values were used to assess confidence intervals.”

It is not clear to me how you tested the difference in R^2 , and whether this is an appropriate statistical test. The way I understand the procedure is that you estimated two distributions of R^2 , by sampling the data 200 times, and used a t-test to compare the mean R^2 . If this is the case, I don't think this is an appropriate test, as its power depends on the number of sampling (here 200) and not on the sample size. Basically, if you had done 10,000 sampling iterations most t-test between the two distributions would come up significant. Maybe I misunderstood, so can you please clarify how the test was performed?

A more straightforward test would consist in fitting a model with the 2 PRS, and comparing it to a model with only the standard PRS. The likelihood ratio test can be used to compare the two nested models. If you really want confidence intervals around R^2 , I suggest to use bootstrap. I am also aware of this R package, if you are after a parametric approach. [https://www.cell.com/ajhg/pdf/S0002-9297\(23\)00004-6.pdf](https://www.cell.com/ajhg/pdf/S0002-9297(23)00004-6.pdf)

Overall, the fact that you only observe highly significant differences on only a handful of PRS scenarios make me doubt of the validity of the tests.

Response: The Reviewer is correct in the understanding that, in the previous version of our manuscript, we estimated two distributions of R^2 by sampling the data 200 times, and used a t-test to compare the mean R^2 . We agree with the Reviewer that the current test is not optimal. Following the Reviewer's suggestion, we have implemented the likelihood ratio test in the revised manuscript. The results from

this analysis supports the advantage of the pleiotropy-informed PGS (pleioPGS) compared to the standard PGS. The main text was updated accordingly.

Revisions:

Introduction, page 4, line 99

Last, we improve polygenic prediction of **bipolar disorder and ADHD** after conditioning the GWASs of five major psychiatric disorders on our multimodal, multivariate, genetic signal of brain morphology, functional connectivity, and tissue composition.

Methods, page 20, line 622

Per disorder we tested if the PGS with the highest R^2 (MDD pleioPGS/SCZ original PGS with $n_{\text{SNP}} = 50,000$; ADHD/ASD/BD pleioPGS with $n_{\text{SNP}} = 100,000$) was significantly higher than the R^2 of the alternative PGS (MDD original PGS/SCZ pleioPGS with $n_{\text{SNP}} = 50,000$; ADHD/ASD/BD original PGS with $n_{\text{SNP}} = 100,000$). For each disorder a likelihood ratio test was applied to test if a model with both pleioPGS and original PGS provides significantly better prediction compared to the model with only one PGS included.

Results, page 10, line 272

For each disorder a likelihood ratio test was then applied to test if a model with both pleioPGS and original PGS provides significantly better prediction compared to the model with only one PGS included.

We observed a 1.24-fold increase in variance explained ($R^2 = 6.22\%$, $p = 2.28 \times 10^{-15}$) for BD in the TOP sample ($N_{\text{case}} = 463$; $N_{\text{control}} = 1,073$) using the pleioPGS with 100,000 SNPs, compared with the original GWAS-based PGS ($R^2 = 5.02\%$, $p = 2.28 \times 10^{-15}$). The likelihood ratio test that compared the model fit before and after adding BD pleioPGS to the original BD PGS showed that pleioPGS significantly improved the model fit ($\chi^2 (df = 2) = 13.35$, $p = 1.26 \times 10^{-3}$). Also for ADHD, the pleioPGS ($R^2 = 1.47\%$, $p = 4.57 \times 10^{-38}$) performed better than the original PGS ($R^2 = 1.20\%$, $p = 2.62 \times 10^{-31}$), contributing significantly to explaining the variation in ADHD liability ($\chi^2 (df = 2) = 33.54$, $p = 5.21 \times 10^{-8}$) in MoBa cases ($N = 2,216$) and controls ($N = 206,644$). In MDD cases ($N_{\text{case}} = 135$) and controls ($N_{\text{control}} = 1,073$), the pleioPGS ($R^2 = 3.98\%$, $p = 2.16 \times 10^{-6}$) explained more variance than the original GWAS-based PGS ($R^2 = 3.22\%$, $p = 1.92 \times 10^{-5}$) as well. Adding MDD pleioPGS to the original MDD PGS did not significantly improve the model fit ($\chi^2 (df = 2) = 5.30$, $p = 0.07$), but pleioPGS remained significant in the combined model while the original MDD PGS became insignificant (original PGS $t = 0.90$, $p = 0.37$; pleioPGS $t = 2.29$, $p = 0.02$). A similar observation was made for ASD: although the higher variance explained by pleioPGS ($R^2 = 1.47\%$, $p = 4.57 \times 10^{-38}$) compared to the original PGS ($R^2 = 1.47\%$, $p = 4.57 \times 10^{-38}$) in BUPGEN ($N_{\text{case}} = 331$, $N_{\text{control}} = 1073$) did not cause a significantly increased model fit ($\chi^2 (df = 2) = 5.93$, $p = 5.17 \times 10^{-2}$), pleioPGS remained significant in the combined model while the original PGS became insignificant (original PGS $t = -0.70$, $p = 0.48$; pleioPGS $t = 2.42$, $p = 15.37 \times 10^{-3}$). When predicting SCZ in TOP ($N_{\text{case}} = 735$; $N_{\text{control}} = 1,073$), the addition of the original GWAS-based PGS provided significant improvement ($\chi^2 (df = 2) = 23.17$, $p = 9.30 \times 10^{-6}$).

Figure 3. Phenotypic variance explained for five psychiatric disorders by polygenic scores based on original disorder GWAS summary statistics (original PGS) and conditional disorder-multimodal GWAS summary statistics (pleioPGS). Note that the PGS using the top 10 and top 100 SNPs had too low R^2 values to be visible and are therefore not plotted. These values can be found in Supplementary Data 17. Independent target samples were used for prediction (see Methods). A likelihood ratio test was applied to compare the model which includes only the best PGS (highest R^2) with the model including both PGS for each disorder.

Discussion, page 12, line 310

Moreover, we showed how these results can be leveraged to improve polygenic prediction of bipolar disorder and ADHD.

Discussion, page 13, line 361

Conditioning GWAS summary statistics for five major psychiatric disorders on our multimodal multivariate analysis prioritized variants that enhanced polygenic prediction for BD and ADHD. This highlights that future diagnostic strategies targeting psychiatric disorders may benefit from aligning the PGSs to a relevant endophenotype. This rationale is similar to a previously published study that showed improved prediction of SCZ after filtering on genetic variants expressed in the placenta¹. Exploring which disorders benefit from either a more specific or broader range of neuroimaging phenotypes could also be an intriguing area for future research.

Minor:

In your point about power of GWAS: Can you clarify what you mean by additive genetic variance, is it the 32% of the SNP heritability or 32% of the twin based heritability (thus close to the 50% mark that is tagged by common SNPs).

“For example, researchers expect that multivariate analyses of the prospective UK Biobank neuroimaging sample of 100,000 individuals will explain 32% and 24% of the additive genetic variance in cortical surface area and thickness respectively”

Response: We thank the reviewer for the opportunity to clarify this statement. The cited percentages are estimated with MiXeR in the original publication², which can be used to describe the proportion of additive genetic variance explained by genome-wide significant SNPs as a function of sample size. Here, additive genetic variance refers to SNP-based heritability estimated by MiXeR³. We have altered the statement in the discussion to clarify this issue.

Revision:

Discussion, page 13, line 340

For example, researchers expect that genome-wide significant variants in the multivariate analyses of the prospective UK Biobank neuroimaging sample of 100,000 individuals will explain 32% and 24% of the SNP-based heritability of cortical surface area and thickness, respectively.

Reviewer #2

The authors have conducted a careful revision of the manuscript. I only have two remaining questions regarding the new Figure 3:

1) The number of loci discovered was used as the criterion comparing the power of univariate and multivariate analysis, but different thresholds were applied: $p < 5e-8$ for univariate and $FDR < 0.05$ for multivariate. Is this really a fair comparison of the statistical power of the two testing strategies?

Response: We fully agree that this is a relevant comment. We used the standard ways of reporting significant findings for the considered analyses to keep consistency and facilitate comparison with previously reported results. However, we agree that these criteria are not directly comparable, as discussed in the condFDR methods descriptions⁴⁻⁶. We have now clarified this limitation in the revised manuscript, and removed the direct comparison from the Results and Discussion since it's not part of the primary analyses and may unnecessarily distract reader's attention from the main message of the study.

Revisions:

Results, page 10, line 257

Alterations in brain morphology, connectivity, and tissue composition often co-occur in heritable psychiatric disorders⁷, suggesting that our multimodal, multivariate genetic signal may have relevance for the genetics discovery and prediction of psychiatric disorders. It is possible to boost locus discovery and polygenic prediction by re-ranking the test-statistics from a given GWAS based on a genetically related secondary GWAS⁸. We therefore conditioned five major psychiatric disorder GWAS summary statistics on our multimodal MOSTest summary statistics using the conditional false discovery rate approach (condFDR; Supplementary Results)⁴. ~~to identify loci associated with~~ We included schizophrenia (SCZ)⁹, bipolar disorder (BD)¹⁰, major depressive disorder (MDD)^{11,12}, attention-deficit hyperactivity disorder (ADHD)¹³, and autism spectrum disorder (ASD)¹⁴ (Supplementary Data 12). The rationale behind condFDR is that, in the presence of cross-trait enrichment, a variant with strong associations with both traits is more likely to represent a true association¹⁵. These disorder-multimodal condFDR summary statistics were used to construct polygenic scores for the disorders in independent samples using a pleiotropy-informed polygenic scoring method¹⁵ (pleioPGS; see Methods). For that

purpose, we constructed polygenic scores with PRSice2 in three independent clinical datasets (TOP, BUPGEN and MoBa, see Supplementary Data 12) based on the top 10-150,000 original disorder GWAS ordered SNPs and condFDR-based ordered SNPs (Fig. 3). For each disorder a likelihood ratio test was then applied to test if a model with both pleioPGS and original PGS provides significantly better prediction compared to the model with only one PGS included.

Compared to the number of genome-wide significant loci identified in the original GWAS (Fig. 3a), we observed a 5-19 fold increase in locus yield in the condFDR summary statistics (Supplementary Data 13). Note that the locus yield from the original GWAS may be lower than the number presented in the original articles, since we relied on publicly available data (e.g. excluding 23andMe) and exclude some additional cohorts to prevent sample overlap (with e.g. UKB, see Supplementary Results 2). We calculated the sign concordance of the condFDR lead SNPs using independent disorder GWAS summary statistics¹⁶⁻¹⁸ (Supplementary Data 12) and observed higher and more significant sign concordance across all disorders compared to the lead SNPs identified in the original GWAS (Supplementary Data 14). We positionally mapped condFDR loci to genes (528 genes for MDD, 2,416 for SCZ, 116 for ASD, 1,162 for BD, 122 for ADHD; Supplementary Data 15) and subsequently used these genes as input for hypergeometric test based tissue enrichment analyses, both as implemented in FUMA. The SCZ and BD genes were significantly enriched in gene sets with upregulated gene expression in brain tissue, but not in other body tissues (Supplementary Data 16, Fig. 3c). This contrasts with the sets of genes mapped from the original GWAS loci (60 genes for MDD, 498 for SCZ, 5 for ASD, 186 for BD, 14 for ADHD; Supplementary Data 15), which showed no enrichment of upregulated gene expression in brain tissues. This illustrates that genetic overlap may be used to identify genetic variants that may play a role in psychiatric disorders with a specific relevance for brain traits.

We observed a 1.24-fold increase in variance explained ($R^2 = 6.22\%$, $p = 2.28 \times 10^{-15}$) for BD in the TOP sample ($N_{\text{case}} = 463$; $N_{\text{control}} = 1,073$) using the pleioPGS with 100,000 SNPs, compared with the original GWAS-based PGS ($R^2 = 5.02\%$, $p = 2.28 \times 10^{-15}$). We used a likelihood ratio test to compare the model fit before and after adding BD pleioPGS to the original BD PGS and show that pleioPGS brings a significant improvement in model fit ($\chi^2 (df = 2) = 13.35$, $p = 1.26 \times 10^{-3}$). Also for ADHD, the pleioPGS ($R^2 = 1.47\%$, $p = 4.57 \times 10^{-38}$) performed better than the original PGS ($R^2 = 1.20\%$, $p = 2.62 \times 10^{-31}$), contributing significantly to explaining the variation in ADHD liability ($\chi^2 (df = 2) = 33.54$, $p = 5.21 \times 10^{-8}$) in MoBa cases ($N = 2,216$) and controls ($N = 206,644$). In MDD cases ($N_{\text{case}} = 135$) and controls ($N_{\text{control}} = 1,073$), the pleioPGS ($R^2 = 3.98\%$, $p = 2.16 \times 10^{-6}$) explained more variance than the original GWAS-based PGS ($R^2 = 3.22\%$, $p = 1.92 \times 10^{-5}$) as well. Adding MDD pleioPGS to the original MDD PGS did not improve the model fit ($\chi^2 (df = 2) = 5.30$, $p = 0.07$), but pleioPGS remained significant in the combined model while the original MDD PGS became insignificant (original PGS $t = 0.90$, $p = 0.37$; pleioPGS $t = 2.29$, $p = 0.02$). A similar observation was made in the case of ASD: although the higher variance explained by pleioPGS ($R^2 = 1.47\%$, $p = 4.57 \times 10^{-38}$) compared to the original PGS ($R^2 = 1.47\%$, $p = 4.57 \times 10^{-38}$) in BUPGEN ($N_{\text{case}} = 331$, $N_{\text{control}} = 1073$) did not cause a significantly increased model fit ($\chi^2 (df = 2) = 5.93$, $p = 5.17 \times 10^{-2}$), pleioPGS remained significant in the combined model while the original PGS became insignificant (original PGS $t = -0.70$, $p = 0.48$; pleioPGS $t = 2.42$, $p = 15.37 \times 10^{-3}$). When predicting SCZ in TOP ($N_{\text{case}} = 735$; $N_{\text{control}} = 1,073$), the addition of the original GWAS-based PGS provided significant improvement ($\chi^2 (df = 2) = 23.17$, $p = 9.30 \times 10^{-6}$).

Figure 3. Phenotypic variance explained for five psychiatric disorders by polygenic scores based on original disorder GWAS summary statistics (original PGS) and conditional disorder-multimodal GWAS summary statistics (pleioPGS). Note that the PGS using the top 10 and top 100 SNPs had too low R^2 values to be visible and are therefore not plotted. These values can be found in Supplementary Data 17. Independent target samples were used for prediction (see Methods). A likelihood ratio test was applied to compare the model which includes only the best PGS (highest R^2) with the model including both PGS for each disorder.

Discussion, page 13, line 361

Conditioning GWAS summary statistics for five major psychiatric disorders on our multimodal multivariate analysis prioritized variants that enhanced polygenic prediction for BD and ADHD. ~~We discovered new loci including genes with increased brain-specific gene expression. The validity of these boosted loci needs to be replicated in independent samples in future research.~~ This highlights that future diagnostic strategies targeting psychiatric disorders may benefit from aligning the PGSs to a relevant endophenotype. This rationale is similar to a previously published study that showed improved prediction of SCZ after filtering on genetic variants expressed in the placenta¹. Exploring which disorders benefit from either a more specific or broader range of neuroimaging phenotypes could also be an intriguing area for future research.

Supplementary Methods, page 4, line 106

We defined genome-wide significant loci (see *Locus definition*) for the original GWAS as $p < 5 \times 10^{-8}$ and conditioned summary statistics as $FDR < 0.05$. ~~We used the standard ways of reporting significant findings for considered analyses to keep consistency and facilitate comparison with previously reported results. However, we acknowledge that these criteria are not directly comparable, as discussed in the condFDR methods descriptions⁴⁻⁶.~~

Supplementary Results, page 6, line 170

The genome-wide significant loci identified in the original GWAS and in the condFDR summary statistics are displayed in Supplementary Data 13. Note that the number of significant loci from the original GWAS may be lower than the number presented in the original articles, since we relied on publicly available data (e.g. excluding 23andMe) and exclude some additional cohorts to prevent sample overlap (with e.g. UKB, see Supplementary Results 2). The sign concordance of the original lead SNPs and condFDR lead SNPs using independent disorder GWAS summary statistics^{16–18} can be found in Supplementary Data 14. We positionally mapped condFDR loci to genes (528 genes for MDD, 2,416 for SCZ, 116 for ASD, 1,162 for BD, 122 for ADHD; Supplementary Data 15) and subsequently used these genes as input for hypergeometric test-based tissue enrichment analyses, both as implemented in FUMA. The same test was applied on genes mapped from the original GWAS loci (60 genes for MDD, 498 for SCZ, 5 for ASD, 186 for BD, 14 for ADHD; Supplementary Data 15). Supplemental Fig. 9 shows the results of the enrichment test for gene-sets with upregulated gene expression in brain and bodily tissues.

Supplementary Figure 9. Enrichment of disorder genes (positionally mapped from the original disorder GWAS loci vs conditional FDR loci) in tissue-specific gene-sets with upregulated expression (Supplementary Data 16). The red line indicates Bonferroni correction for the number of tests ($\alpha = 0.05/540 = 9.26 \times 10^{-5}$).

2) The text in the figure color keys looks funny. I'm not sure what went wrong, but it also looks so in the rebuttal letter.

Response: We thank the Reviewer for pointing out this error. We have resolved the issue in the revised manuscript.

References

1. Ursini, G. *et al.* Convergence of placenta biology and genetic risk for schizophrenia. *Nat. Med.* **24**, 792–801 (2018).
2. Shadrin, A. A. *et al.* Vertex-wise multivariate genome-wide association study identifies 780 unique genetic loci associated with cortical morphology. *Neuroimage* **244**, (2021).
3. Frei, O. *et al.* Bivariate causal mixture model quantifies polygenic overlap between complex traits beyond genetic correlation. *Nat. Commun.* **10**, 1–11 (2019).
4. Andreassen, O. A. *et al.* Improved Detection of Common Variants Associated with Schizophrenia and Bipolar Disorder Using Pleiotropy-Informed Conditional False Discovery Rate. *PLoS Genet.* **9**, (2013).
5. Smeland, O. B. *et al.* Discovery of shared genomic loci using the conditional false discovery rate approach. *Hum. Genet.* **139**, 85–94 (2020).
6. Thompson, W. K. *et al.* An Empirical Bayes Mixture Model for Effect Size Distributions in Genome-Wide Association Studies. *PLoS Genet.* **11**, 1–21 (2015).
7. Liu, S. *et al.* Multimodal neuroimaging computing: a review of the applications in neuropsychiatric disorders. *Brain Informatics* **2**, 167–180 (2015).
8. Hindley, G. *et al.* Multivariate genetic analysis of personality and cognitive traits reveals abundant pleiotropy. *Nat. Hum. Behav.* (2023).
9. Trubetskoy, V. *et al.* Mapping genomic loci implicates genes and synaptic biology in schizophrenia. *Nature* **604**, 502–508 (2022).
10. Mullins, N. *et al.* Genome-wide association study of over 40,000 bipolar disorder cases provides new insights into the underlying biology. *Nat. Genet.* **53**, 817–829 (2021).
11. Wray, N. R. *et al.* Genome-wide association analyses identify 44 risk variants and refine the genetic architecture of major depression. *Nat. Genet.* **50**, 668–681 (2018).
12. Levey, D. F. *et al.* Reproducible Genetic Risk Loci for Anxiety: Results From ~200,000 Participants in the Million Veteran Program. *Am. J. Psychiatry* **177**, 223–232 (2020).
13. Demontis, D. *et al.* Discovery of the first genome-wide significant risk loci for attention deficit/hyperactivity disorder. *Nat. Genet.* **51**, 63–75 (2019).
14. Grove, J. *et al.* Identification of common genetic risk variants for autism spectrum disorder. *Nat. Genet.* **51**, 431–444 (2019).
15. van der Meer, D. *et al.* Boosting Schizophrenia Genetics by Utilizing Genetic Overlap With Brain Morphology. *Biol. Psychiatry* **92**, 291–298 (2022).
16. Middeldorp, C. M. *et al.* A Genome-Wide Association Meta-Analysis of Attention-Deficit/Hyperactivity Disorder Symptoms in Population-Based Pediatric Cohorts. *J. Am. Acad. Child Adolesc. Psychiatry* **55**, 896–905 (2016).
17. Kurki, M. I., Karjalainen, J., Palta, P., Sipilä, T. P. & Kristiansson, K. FinnGen: Unique genetic insights from combining isolated population and national health register data. *medRxiv* (2022).
18. Lam, M. *et al.* Comparative Genetic Architectures of Schizophrenia in East Asian and European Populations. *Nat. Genet.* **29**, (2019).

Reviewer #1 (Remarks to the Author):

The authors have addressed all my comments

Reviewer #1 (Remarks on code availability):

Yes, code looks great. The authors provide code to generate and clean the phenotypes, perform the main analyses and generate the figure of the article. The code is well documented.

Reviewer #2 (Remarks to the Author):

Thanks for the revision. Regarding the software, the link works but the documentation is limited. The page points to a pleioFDR page only if I go into the pleioPRS folder. I would suggest the authors clearly state what MOSTest-multimodal is, how it is related to MOSTest and pleioFDR/pleioPRS, and directly give installation instructions with step-by-step real data examples in the readme of the MOSTest-multimodal GitHub page.

Reviewer #2 (Remarks on code availability):

I apologize that I do not have time to really test whether the code provided can reproduce the entire reported results. The link works but the documentation is limited. The page points to a pleioFDR page only if I go into the pleioPRS folder. I would suggest the authors clearly state what MOSTest-multimodal is, how it is related to MOSTest and pleioFDR/pleioPRS, and directly give installation instructions with step-by-step real data examples in the readme of the MOSTest-multimodal GitHub page.

Please find below our response to the remaining comments raised by the reviewers together with the adjustments we made to the manuscript. The questions and comments of the reviewer are presented in italics, our response in normal typeset. We marked changes to the text in blue in the updated version of our manuscript.

Reviewer #1

The authors have addressed all my comments. Yes, code looks great. The authors provide code to generate and clean the phenotypes, perform the main analyses and generate the figure of the article. The code is well documented.

Response: We thank the reviewer for their positive evaluation.

Reviewer #2

Thanks for the revision. Regarding the software, the link works but the documentation is limited. I apologize that I do not have time to really test whether the code provided can reproduce the entire reported results. The page points to a pleioFDR page only if I go into the pleioPRS folder. I would suggest the authors clearly state what MOSTest-multimodal is, how it is related to MOSTest and pleioFDR/pleioPRS, and directly give installation instructions with step-by-step real data examples in the readme of the MOSTest-multimodal GitHub page.

Response: We are happy to elaborate on the MOSTest-multimodal GitHub repository. It is important to clarify that we did not develop new software tools. In this manuscript, we have used different openly available existing tools (MOSTest, pleioFDR, PRSice2 etc.) that have been published elsewhere. What we provide in our repository is the analysis code wherein we use the tools, the code we use to glue them together in a pipeline, and finally produce Figures of the results.

We have now elaborated the readme files on our GitHub repository to make this clearer. As each of these tools that we used have their own installation and user guides, either on GitHub or on other platforms, we have now described these in more detail and included the references. See some examples below:

README.md ✎

SNP-based heritability and genetic correlations were estimated for each phenotype (combination) using Linkage Disequilibrium Score Regression (LDSC; Bulik-Sullivan et al., 2015).

For installation and step-by-step user guides, please refer to the repository of the software on <https://github.com/bulik/ldsc>.

In short:

1. Munge your summary statistics with `munge.slurm.sh` to transform them into the standard LDSC format.
2. Use the munged summary statistics to run `h2.slurm.sh`, this will give you the SNP-based heritability estimates. Plot the results with `plot_ldsc_h2_results.R`
3. Use the munged summary statistics to run `rg.slurm.sh` and obtain all pairwise genetic correlations between the traits. Plot the results with `plot_ldsc_rg_results.ipynb`

Outline: association z-scores from non-permuted GWAS are used to estimate MOSTest test statistics, while z-scores from the permuted GWAS are used to estimate distribution of z-scores under null assumption of no association (null distribution). Null distribution is then used to calculate p-values of the the MOSTest test statistics. Produced MOSTest sumstats are then clumped to produce independent loci and corresponding lead SNPs. Finally Manhattan plot is produced.

1. Create z-matrices with original and permuted GWAS z-scores across all phenotypes with `mostest_multimodal.ipynb`.
 - Two NxM matrices are created: one for original genotypes and another for permuted genotypes, where N is the number of variants tested in the GWASs and M is the number of phenotypes.
 - Each matrix is saved as three separate files with specified common prefix: `<prefix>.cols` file - a single column with variant IDs, `<prefix>.rows` file - a single column with phenotype names, `<prefix>.dat` - binary file with NxM matrix of z scores (in float32 format).
 - NB! the order of variants and phenotypes should be the same in the original and corresponding permuted files. `<prefix>.dat` files are then used in the `mostest_multimodal.m` to estimate mostest test statistics and corresponding p-values.
2. Run MOSTest with `submit_mostest.slurm`. The main input of MOSTest are
 - univariate GWAS summary statistics for each phenotype based on original genotypes (see `plink` folder)
 - univariate GWAS summary statistics for each phenotype based on permuted genotypes (see `plink` folder)
 - z-matrices created in 1.
3. With `submit_pvals2csv.slurm`:
 - Convert matlab output to CSV file
4. With `submit_clump.slurm`:
 - Create "FUMA-style" genome-wide significant loci, using https://github.com/precimed/python_convert
5. With `Manhattan_plots.R`:
 - Create Manhattan plot for each multivariate GWAS output

Outline: run univariate GWAS for all phenotypes for both original (non-permuted) genotypes and for randomly permuted genotypes. Genotypes are split into chunks to allow parallel processing, providing considerable speedup when running on the HPC.

1. Produce dataset of unrelated individuals and remove rare (MAF<0.005) SNPs with `remove_related.sh`. This dataset will be used for all subsequent analyses.
2. Split list of all SNP IDs into chunks with 10000 SNPs (resulting in 907 chunks) with `make_snp_chunks.sh`.
3. Use `make_bfile_chunks.slurm.sh` to
 - split genotypes into chunks using SNP chunks generated in step 2, i.e. 907 (non-permuted) genotype datasets are produced each containing all individuals and 10000 SNPs (the last chunk contains less SNPs).
 - make chunks of permuted genotypes randomly permuting each non-permuted genotype chunk (`permute_bed.py` python script is used to generate permuted genotypes).
4. Use `merge_chunks.sh` to merge all chunks of permuted genotypes.
5. Run univariate GWAS for the original (non-permuted) genotypes (using `gwas_chr.slurm.sh` and genotypes produced in step 1) and randomly permuted genotypes (using `gwas_perm_chr.slurm.sh` and genotypes produced at step 4). For computational efficiency GWASs are run per-chromosome.
6. Use `make_LD_ref.sh` to make genotype files which are subsequently used as an LD reference for clumping of the MOSTest sumstats. `LD_ref.txt` file contains a list of 1000 individuals randomly selected from the dataset obtained in step 1.

After phenotype definition (`MOSTest-multimodal/phenotypes/`), running all univariate GWAS (`MOSTest-multimodal/plink/`), and combining them into multivariate summary statistics (`MOSTest-multimodal/mostest/`), we were interested in the following analyses:

1. Locus overlap between the three MOSTest summary statistics (sMRI, fMRI, dMRI) was defined as physically overlapping genome-wide significant loci. Herefore we used the GenomicRanges R-package, for more information see <https://github.com/Bioconductor/GenomicRanges>. In short, this compares the chromosome and start and end base pair positions of all loci between any pair of summary statistics as done in `Loci_overlap_GRanges.R`.
2. Also the genes that were found to be genome-wide significant in MAGMA [ran on <https://fuma.ctglab.nl>] were compared to provide a similar overview. The overlapping patterns were then plotted with the eulerr R-package, see `Gene_overlap.R`.
3. Functional consequences of lead SNPs were provided by ANNOVAR as implemented in FUMA [<https://fuma.ctglab.nl>] and then compared to Watanabe et al. 2019 (Nature Communications) to test for enrichment in `ANNOVAR_enrichment.R`.
4. We used Gene Ontology gene-sets to test for enrichment of genes using hypergeometric testing as implemented in FUMA [<https://fuma.ctglab.nl>]. The exported significant Gene Ontology terms were visualized as a graph using Cytoscape, EnrichmentMap and AutoAnnotate following the Nature Protocol by Reimand & Isserlin et al. and available in `Cytoscape_prep.R`. For more information on how to make Cytoscape network graphics, see: <https://github.com/reimandlab/ActivePathways>.
5. We visualized the temporal gene expression pattern across development using data from Kang et al [<http://www.humanbraintranscriptome.org>] using the code in `GeneExpression_simpleMean.R`.
6. We explored cell-type enrichment of genes by performing Fisher Exact Tests for each cell-type identified by Bhaduri et al. using single-cell RNA sequencing analyses of the fetal brain in `MOSTest_CellTypeEnrichment.R`, of which the results were then plotted in `plot_celltype_enrichment.R`.
7. For Supplementary Figure 9, we used FUMA [<https://fuma.ctglab.nl>] to test for the enrichment of disorder genes (positionally mapped from the original disorder GWAS loci vs conditional FDR loci) in tissue-specific gene-sets with upregulated expression and visualized the results with `Plot_tissue_specificity.R`.